# Gathering Southward under Secularization and Syncretism: Study of the Spatial-Temporal Distribution and Influencing Factors of Chinese Historical Buddhist Architecture in Zhejiang

**Fei Ju**

College of Art Design, Nanjing Forestry University, Nanjing 210037, China; jufei1986@njfu.edu.cn

**Abstract:** After Buddhism spread to the Zhejiang, it underwent sinicization, giving rise to Chinese Buddhist architecture and forming a secularized character. The spatial-temporal distribution of historical Buddhist architecture in Zhejiang is strongly representative of Buddhist architecture. From the perspective of religious cultural geography, this study takes 11 prefecture-level cities of Zhejiang as the basic research units, and employs the spatial-analysis method of ArcGIS to analyze the spatial-temporal evolution characteristics of representative historical Buddhist architectural samples, and to explore the factors affecting their distribution. The research results indicate that the spatial distribution of representative historical Buddhist architectural samples in the Zhejiang is extensive, with a distinct single-core clustering characteristic. The mean-center movement of the representative historical Buddhist architectural samples in Zhejiang during different historical periods manifests in four distinct directional phases, overall moving from north to south. Natural, transportation, political, technological, economic, and religious cultural transmission factors collectively influence the spatial-temporal distribution of Buddhist architecture in Zhejiang. Especially the secularization of Buddhism and the syncretism of Confucianism, Daoism, and Buddhism have been the primary drivers in the spatial-temporal distribution evolution of Buddhist architecture.

**Keywords:** secularization trend; Chinese historical Buddhist architecture; spatial-temporal distribution; influencing factors

## 1. Introduction

### 1.1. The Evolution Trend of Buddhist Culture in Zhejiang

Buddhism originated in ancient India and was introduced to China as a foreign culture during the Eastern Han Dynasty (25–220). Buddhism gradually integrated into Chinese tradition, becoming one of the three major religions alongside Confucianism and Daoism. Throughout its dissemination, Buddhism experienced a process of sinicization and secularization, profoundly influencing the mainstream strata of Chinese feudal society.

Buddhism was introduced to Zhejiang towards the end of the Eastern Han dynasty (25–220), initially centered in Kuaiji 会稽[1], notably around Mount Kuaiji in Shaoxing 绍兴, Zhejiang. The dissemination of Buddhist scriptures by the Parthian Prince An Shigao 安世高 marked an initial phase of Buddhism's spread in the Zhejiang (Shi 1992, p. 6). Subsequently, due to the frequent warfare in the northern region and frequent exchanges along both the Silk Road and the Maritime Silk Road, there was a facilitation of the migration of monks from northern China, various Western Regions, and South Asia to Zhejiang. These factors jointly promoted the further spread of Mahayana Buddhist scriptures in Zhejiang. Historical records note that the Yuezhi 月氏 individual Zhi Qian 支谦 came to the Eastern Wu (222–280) in the late Eastern Han Dynasty, translated 49 scriptures, and was conferred the title of a scholar by the founding emperor of Eastern Wu, Sun Quan (Shi 1992, p. 15).

The translation and dissemination of Buddhist scriptures facilitated the dissemination of Indian Buddhist culture in Zhejiang. In the process, it continuously conflicted and integrated with Chinese society, culture, and values. Initially, Buddhist culture assimilated

elements of Daoism, which was predominant during the Han and Jin dynasties (25–420). After the Southern Dynasties (420–589), Buddhism was profoundly influenced and Confucianized by Confucian culture. Starting from the Song dynasty (920–1279), Buddhism entered a mature phase of sinicization. Influenced by the converging trends of Confucianism, Buddhism, and Daoism, Buddhist culture integrated deeply with traditional folk culture, showcasing a prominent trend towards secularization. Buddhism not only integrated with Chinese culture and values, creating a distinct Chinese Buddhist culture, but also broadened its scope from purely religious aspects to other realms of Chinese society, influenced by political and economic factors.

The secularization of Buddhism in China is a dynamic process of continuous self-adjustment, seeking to align more closely with the secular society (Chen 2019, pp. 157–63). This evolution is manifested on several fronts. Firstly, the secularization of Buddhism is exhibited through the nationalized management of Buddhist temples, which have become religious organizations regulated by government institutions. Secondly, Buddhist temples and monks actively engage in social services, offering educational and charitable assistance to the common folk, further deepening the secularization of Buddhism. Thirdly, Buddhism has become deeply entrenched within China's folk–belief system, often intertwining with traditional faiths. In some folk Buddhist temples, Buddha statues are worshipped alongside Daoist deities or folk deities. Furthermore, Buddhist moral principles are frequently applied in social life, illustrating the convergence of Buddhist ceremonial activities with the commercial economy, a phenomenon visibly seen in the business activities surrounding the temples.

Overall, the sinicization of Buddhism has spurred the evolution of Buddhist culture in the realms of Chinese societal thought and culture, with secularization vividly manifested in the specific social practices and applications of Buddhist culture. In Zhejiang, the ruling class significantly backed this evolution, endorsing the construction of Buddhist architectural projects and supporting the monastic activities. Mainstream society's acceptance and reverence of Buddhist culture, along with the assimilation of local beliefs, rituals, and customs, has reciprocally advanced both the sinicization and secularization of Buddhism. Consequently, Buddhism in Zhejiang has presented diverse sects and unique styles, expanding its influence substantially and fully realizing a sinicized transformation across doctrines, rituals, architecture, and institutional frameworks.

### 1.2. The Secularization Trend in the Development of Historical Buddhist Architecture in Zhejiang

Buddhist architecture, emerging from the confluence of Buddhist culture and belief, represents a concrete manifestation of Buddhist culture. This specific type of religious architecture evolved gradually on the foundation of pre-existing Chinese architectural styles. Buddhist architecture serves as an important place for housing Buddha statues, scriptures, and facilitating the situating of monastic communities, as well as for propagating Buddhist teachings, scholarly engagements, ritual congregations, and facilitating religious experiences (Li 2017, pp. 48–49; Zuo 2014, pp. 60–63+226). Common architectural forms within Buddhism encompass temples, pagodas, grottos, and Buddhist sutra pillars.

In the early phases, Buddhist architecture was known by three predominant naming conventions. The first was derived from the ancient Chinese governmental term Honglu Si 鸿胪寺, generally referred to as Si 嗣 (Liu 2020, p. 78; Dong 1983, p. 7498), which was a transformation initiated by the initial confusion between Buddhist architecture and the indigenous Daoist and Confucian ritual places. The second nomenclature, Jialan 伽蓝 (Zhang 1997, p. 18), was derived from the transliteration of the Sanskrit term "Saṃghārāma", which was used in Indian Buddhist architectural nomenclature, retaining its usage up until the Tang Dynasty (618–907). The third nomenclature, "Futu 浮屠 Pagoda," originated from the transliteration of the Sanskrit term "stūpa", which refers to the Indian Buddhist tower–like structures found in early Buddhist architecture. During the Southern Dynasties (420–589), Buddhist temples were commonly referred to as Ta temples 塔寺 (Xu 1986, p. 22). How-

ever, after the Tang Dynasty (618–907), as Buddhism flourished, the term Buddhist temple 佛寺 gradually replaced Jialan 伽蓝 and became the most commonly used term for Buddhist architecture among mainstream society, as was extensively documented in Buddhist scriptures translated by monks.

The White Horse Temple 白马寺, situated in Luoyang and constructed during the Han Dynasty, represents the inception of Buddhist architecture in China (Wei 1974, p. 3029), mentioned in the *Wei Shu·Shi Lao Zhi* 魏书·释老志 (Wei 1974, p. 3029). In Zhejiang, the earliest Buddhist architectural example is the Jinsu 金粟 Temple constructed by Kang Senghui 康僧会[2] during the Eastern Wu period (229–280) in Haiyan 海盐 (Song 2014, p. 1494). From the Western Jin Dynasty (265–317) onward, Zhejiang emerged as a pivotal locus for Buddhist culture in China. The promotion by ruling classes, the desire of mainstream society to seek relief from true suffering through religion, and the southward migration of Buddhist scholars collectively facilitated the widespread dissemination of Buddhism in Zhejiang (Ren 1985, p. 46). According to existing historical records, there was a total of 56 temples built in Zhejiang during the Jin Dynasty (Dan 2022, pp. 40–49, 156).

Beginning in the Southern Dynasties period (420–589), Zhejiang has emerged as a gathering place for Buddhist intellectuals, attributed to its advantageous geographical environment, a relatively stable social environment compared to the continuously war-torn northern regions, and a more developed economy. This engendered the development of a regional Buddhist center, centralized around Kuaiji 会稽. The ruling classes and scholar officials predominantly revered Buddhism, with Emperor Wu of Liang Dynasty (464–549) even earning the epithet "Emperor Bodhisattva" (Yang 2010, p. 233). This period saw a substantial promotion of the construction of Buddhist temples across the province, totaling 132 temples, which far surpassed the combined total from previous generations. The Liang Dynasty (502–557), in particular, witnessed the highest number of constructions, accounting for 95 temples (Mo 1994, pp. 93–96).

During the Sui (581–618) and Tang (618–907) dynasties, Buddhism permeated more deeply into the society of Zhejiang. The reverence for Buddhism held by the ruling class, combined with the establishment of China's earliest Buddhist sect, the Tiantai 天台 Sect, propelled Buddhism into a mature phase of sinicization, culminating in a landscape where multiple sects coexisted. The economic prosperity of the Tang dynasty further influenced the regional distribution of Buddhist architecture, as well as the emergence of a hierarchical differentiation in the architectural presentations that paralleled the social strata. During the Five Dynasties and Ten Kingdoms period (907–979), Hangzhou rose as the capital of the Wuyue 吴越 Kingdom (907–978), and emerged as the only economically prosperous region in China, untouched by warfare. The extensive expansion and establishment of numerous temples earned Hangzhou the title of "Southeast Buddhist Kingdom" (Zhai 2016, p. 69). Consequently, it emerged as the epicenter of Buddhism in Zhejiang and the entire Jiangnan region.

The Song Dynasty marked the pinnacle of Buddhist development in Zhejiang, with the ruling class continuing the policy of extensive Buddhist temple construction initiated in earlier periods. This led to a proliferation of temples, with the number in Hangzhou alone skyrocketing to 360 during the Northern Song Dynasty (Su 1982, p. 644). The grotto statue at Feilai 飞来 Peak[3] in Lingyin 灵隐 Temple, in Hangzhou 杭州, represented the greatest achievement of Buddhist art in Zhejiang during the Song Dynasty.

After the Yuan Dynasty (1271–1368), Tibetan Buddhism prevailed among the ruling class, and the Linji school of Chan 禅宗 Buddhism emerged as the mainstream sect of Chinese Buddhism (Zhang 2002, p. 3). Its focal point of activity, Tianmu 天目 Mountain also became the center for Chinese Buddhism in Zhejiang. Following the Ming dynasty, Chinese Buddhism gradually secularized, evolving more into a folk custom and fostering a distinct "Jiangdong Buddhist style" (Chen 2001, p. 4). During the Qing dynasty (1616–1912), Tibetan Buddhism held sway among the ruling class, leading to a further decline in Chinese Buddhism in Zhejiang, with only the Pure Land 净土 School prevailing as the mainstream belief among the populace. With the changing social atmosphere during the late Qing

Dynasty and the Republican era (1912–1949), Buddhist education initiatives surged, characterized by the significant emergence of Buddhist academies and organizations, marking an important feature of the development of Buddhism in the Zhejiang during this period.

*1.3. Research Methods of the Spatiotemporal Distribution of Historical Buddhist Architecture under the Influence of Religious Geography*

Given that there are few phenomena that are as latent and sensitive as religion in forming and reflecting cultural regional differences (Zelinsky 1961, pp. 139–93), religious geography has accordingly become a significant focus within the broader discipline of cultural geography (Xue and Zhu 2010, pp. 89, 109–13). Geographers' attention to the influence of various geographical environments on religious beliefs has cultivated a research relationship between religion and the geographical environment (Jordan-Bychkov and Mona 1999, pp. 215–59), the distribution and diffusion of religion (Park 1994, pp. 1–123), and religious cultural religions (Kent and Neugebauer 2010, pp. 425–41). This interdisciplinary approach has brought innovative perspectives and methods to the study of religion.

The Geographical Information System (GIS) possesses advanced spatial-analysis functions and intuitive visualization effects (Ivakhiv 2006, pp. 169–75), enabling the intuitive representation of the spatial distribution and diffusion of geographical elements. It is widely used in the study of religious geography, assisting scholars in analyzing the spatial-distribution characteristics of religious organizations and sites, comparing regional differences, and analyzing their underlying causes (Bae 2007, pp. 139–51). Enoch Cheng used county-level spatial-statistics methods to analyze the spatial distribution of religious organizations in the United States and their socioeconomic characteristics, discovering that socioeconomic and demographic variables impact the level of spatial clustering of religious organizations (Cheng and Meng 2023, pp. 789–812). Bae Sun Hak analyzed the current state of Buddhist temple sites, finding that various factors such as natural environment, historical background, and geographical location have influenced the spatial distribution of temple sites. (Bae 2007, pp. 139–51).

In studies exploring the spatial distribution of religious spaces in Chinese religious architecture, Yang Fenggang discovered that the secularization of Buddhism in China is distinct from that of religions in Europe (Yang 2016, pp. 19–35). Chen Junzi identified that the socioeconomic environment of different historical periods, governmental religious policies, and the pathways of religious transmission are crucial factors influencing the spatial distribution of religious architectural heritage sites (Chen et al. 2018, pp. 84–90). Zhu Puxuan argued that support from the ruling class was an important condition affecting the diffusion of Tibetan Buddhist culture, and the spatial arrangement of Buddhist temples in Qinghai following the Yuan Dynasty (Zhu 2009, pp. 8–14; Zhu 2010, pp. 37–41+122). Wang Liping validated that the method of religious transmission is one of the important factors influencing the spatial distribution of Tibetan Buddhist temples across different historical periods. (Wang and Zhou 2017, pp. 155–61). Li Xiangyu discovered that regional historical and cultural factors have shaped the spatial distribution of Buddhist architecture (Li and Liang 2012, pp. 176–81+186). Xu Ying pointed out that the spatial distribution of Buddhist architecture during the Sui and Tang periods reflects the influence of the natural environment on the dissemination of Buddhism (Xu and Fang 2013, pp. 4577–81).

Ultimately, historical Buddhist architecture serves as a vessel for the Buddhist culture of different historical periods, and its spatial-temporal distribution can reflect, to a certain extent, the historical trajectory of Buddhist development in China. Currently, both domestically and internationally, there are fewer studies on Chinese historical Buddhism's architecture from a geographical spatial perspective. Although some scholars have studied the distribution of historical Buddhist architecture on a mesoscopic scale for different historical periods and specific regions, these studies primarily focus on Tibetan Buddhism and lack analysis on the spatial-temporal distributions and influencing factors of Chinese historical Buddhist architecture, especially in the Zhejiang region.

Zhejiang is one of the provinces in China with a well-developed Chinese Buddhist culture and a substantial distribution of Buddhist architecture throughout history. This study selects historical Buddhist architecture in Zhejiang as its subject matter. From the perspective of religious cultural geography and through employing spatial-analysis methods of ArcGIS to explore the spatial-temporal-distribution characteristics of representative Chinese historical Buddhist architectural sites in Zhejiang Province, it conducts in-depth research on the spatial distribution of Buddhist architectural sites of different types and from historical periods, aiming to provide a scientific basis for the preservation and utilization of historical Buddhist architecture.

The study reveals that the spatial distribution of historical Buddhist architecture in Zhejiang is widespread, displaying a spatial clustering with a clear trend of single-core clustering. The center of gravity has been migrating from the north to the south since the Three Kingdoms period, with the Sui, Tang, and Song dynasties representing the pinnacle in the development of historical Buddhist architecture. This spatial-temporal evolutionary pattern is deeply influenced by various factors, including natural elements, transportation, political conditions, technological developments, and the diffusion of religious culture. Among them, the secularization of Buddhism and the converging trend of Confucianism, Daoism, and Buddhism are the primary driving forces affecting the morphological evolution of historical Buddhist architecture.

## 2. Data Sources and Research Methods

### 2.1. Data Sources

This study selects 159 representative historical Buddhist architectural samples from the Three Kingdoms to the Republican era, with complete initial-construction-date information from the National Buddhist temples in the Han region[4], 2022 provincial-level sinicized religious sites[5], National Key Cultural Relics Protection Units[6], and Zhejiang provincial key cultural relics protection units[7]. These samples are categorized into six types: Buddhist temples, constituent structures of Buddhist temples (palaces, bridges and pavilions), pagodas, Buddhist sutra pillars, grottos, and statues (Table 1).

**Table 1.** The statistical information of representative historical Buddhist architecture in Zhejiang.

| City | Buddhist Temple | Constituent Structure | Pagoda | Pillar | Grotto | Statue | Total | Proportion |
|---|---|---|---|---|---|---|---|---|
| Hangzhou | 10 | —— | 14 | 3 | —— | 7 | 34 | 0.214 |
| Huzhou | 9 | —— | 3 | —— | —— | —— | 12 | 0.013 |
| Jiaxing | 7 | 1 | 1 | 4 | —— | —— | 13 | 0.082 |
| Jinhua | 8 | 1 | 5 | 1 | —— | —— | 15 | 0.094 |
| Lishui | 5 | —— | 3 | —— | —— | —— | 8 | 0.050 |
| Ningbo | 13 | —— | 1 | —— | 1 | —— | 15 | 0.094 |
| Quzhou | 3 | —— | 2 | —— | —— | —— | 5 | 0.031 |
| Wenzhou | 9 | 2 | 11 | —— | 2 | —— | 24 | 0.151 |
| Zhoushan | 4 | —— | 1 | —— | —— | —— | 5 | 0.031 |
| Shaoshing | 3 | —— | 2 | —— | —— | 3 | 8 | 0.050 |
| Taizhou | 12 | —— | 7 | —— | —— | 1 | 20 | 0.126 |
| Total | 83 | 4 | 50 | 8 | 3 | 11 | 159 | —— |
| Proportion | 0.522 | 0.025 | 0.314 | 0.050 | 0.019 | 0.044 | —— | —— |

### 2.2. Research Methods

Based on the specific addresses of Buddhist architectural samples in Zhejiang, longitude and latitude were extracted using the Amap API coordinate picker via Python, abstracting the samples as points within geographical space. ArcGIS was utilized to establish a spatial-attribute database of the samples and the used data, and used this, in conjunction with data on roads, rivers, and administrative boundaries provided by the National Basic Geographic Information Public Service Platform, as the data sources for plotting the distribution map of the samples. Through statistical-analysis methods, the quantities of samples

from different periods and of different types were analyzed, summarizing the distribution characteristics of historical Buddhist architecture in Zhejiang, and exploring the laws of spatiotemporal evolution and factors influencing their distribution.

### 2.2.1. Kernel Density Estimation Method

Kernel density estimation is a common nonparametric estimation method in spatial point-pattern analysis, serving as a means to visually represent point-distribution patterns. It can be used to investigate the spatial variation in point density within a region and to study the characteristics of point distribution. Using the position of each sample point $i(x, y)$ as the center, calculate the density contribution value of each sample point within a circle with a specified radius $h$, using the kernel function $K$ (). The calculation formula is as follows:

$$\hat{f}(x) = \frac{1}{nh^d} \sum_{i=1}^{n} K\left(\frac{x - x_i}{h}\right) \tag{1}$$

In the formula, $n$ is the number of samples, $d$ is the dimension, and $x - x_i$ represents the distance from the estimated point $x$ to the sample point $x_i$.

### 2.2.2. The Average Nearest-Neighbor Index Method

The average nearest-neighbor index is a crucial metric for spatial geography, describing the distance between the centroid of each element and the centroid position of its nearest-neighbor elements. Calculating the average values of all these nearest-neighbor distances can be used to illustrate and characterize the spatial-distribution type of point-based elements. The calculation formula is as follows:

$$ANN = \frac{\overline{D}_O}{\overline{D}_E} \tag{2}$$

$$\overline{D}_O = \frac{\sum_{i=1}^{n} d_i}{n} \tag{3}$$

$$\overline{D}_E = \frac{0.5}{\sqrt{n^2/A}} \tag{4}$$

In the formula, $\overline{D}_O$ is the average distance between each observed feature and its nearest neighbor; $\overline{D}_E$ indicates the expected average distance for a random distribution of features; $d_i$ equals the distance between the feature $i$ and its nearest-neighboring feature; $n$ corresponds to the total number of features; and $A$ is the area of the minimum bounding rectangle surrounding all features. The z-score value of the average nearest-neighbor index is calculated as follows:

$$Z = \frac{\overline{D}_O - \overline{D}_E}{SE}, \tag{5}$$

$$SE = \frac{0.26136}{\sqrt{n^2/A}} \tag{6}$$

In the return results of the average nearest-neighbor method of spatial statistics, the degree of "clustered" or "dispersed" feature points in spatial distribution needs to be determined by combining ANN values and z-score values. Specifically, when 0 < ANN value < 1, the smaller the value, the more clustered the features are in the space. The z-score is a multiple of the standard deviation, associated with the standard normal distribution, and its value is a measure of statistical significance. When the critical value (z-score) < −2.58, the point-distribution significantly tends towards "clustered". When the critical value (z-score) is between −1.65 and 1.65, it significantly tends towards "random". When the critical value (z-score) > −2.58, the point-distribution significantly tends towards "dispersed".

### 2.2.3. Mean Center Method

The mean center refers to the average *x* coordinate and *y* coordinate of all the elements within a study area. It is extremely beneficial for tracking the trend of changes in the element distribution and comparing the distributions of different types of elements. This method can be used to study Buddhist architecture across different periods and identify the areas where it is concentrated. The calculation formula is provided as follows:

$$\overline{X} = \frac{\sum_{i=1}^{n} x_i}{n}, \overline{Y} = \frac{\sum_{i=1}^{n} y_i}{n} \tag{7}$$

In the formula, *X* and *Y* respectively represent the longitudinal and latitudinal coordinates of the arithmetic mean-center point of the distribution of samples in Zhejiang during different periods. $x_i$ and $y_i$ denote the longitude and latitude of the coordinates of the *i*-th Buddhist architectural sample, and *n* equals the total number of historical Buddhist architectural samples in Zhejiang.

## 3. Spatial-Temporal Pattern and Evolution of Historical Buddhist Architecture in Zhejiang

### 3.1. Spatial-Distribution Characteristics of the Representative Historical Buddhist Architecture in Zhejiang

Under the geographical influences of the integrative river and sea culture, and the connectivity along riverbanks in Zhejiang, most of the most representative historical Buddhist architectural samples are located in areas with dense river networks, primarily in plain regions, while fewer samples are in mountainous and hilly areas (Figure 1).

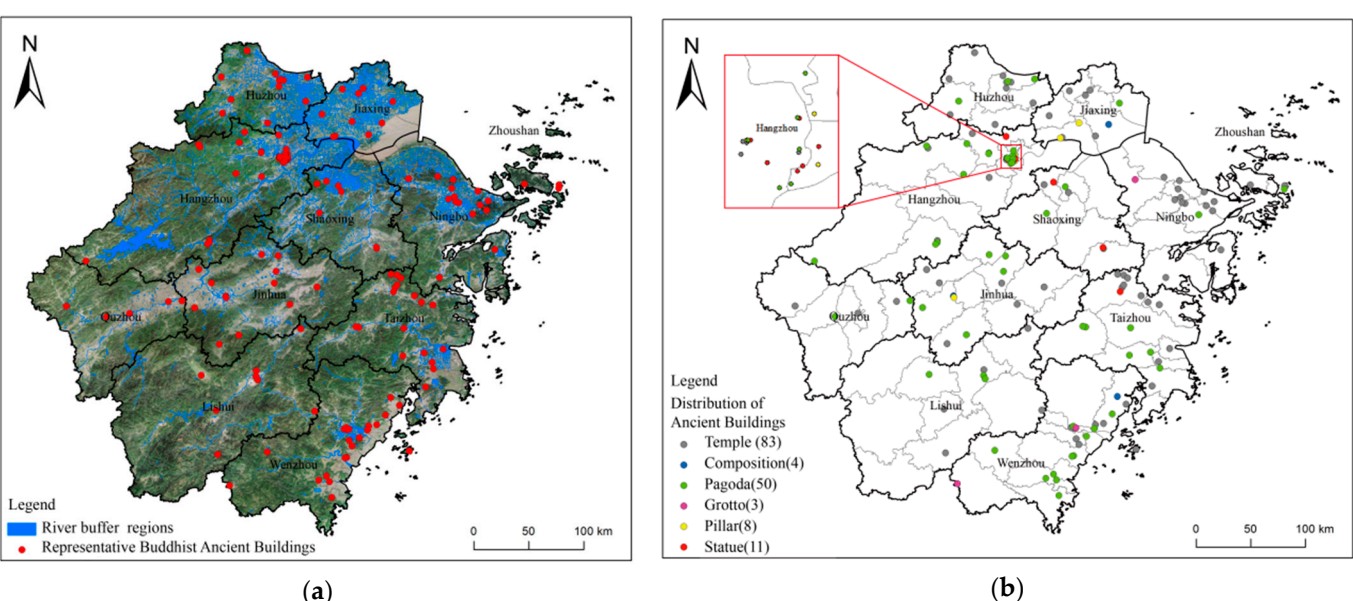

**Figure 1.** Distribution of the representative historical Buddhist architecture samples in Zhejiang Province: (**a**) topographical distribution; (**b**) distribution of samples.

Utilizing GIS spatial analysis, it is calculated that the average nearest-neighbor index (ANN) of representative historical Buddhist architectural samples in Zhejiang is 0.648475 < 1, indicating a clustered distribution. Through constructing a kernel density-distribution map of samples in Zhejiang using ArcGIS (Figure 2), its distribution exhibits a distinct core-edge characteristic. The central urban area in the northeast part of Hangzhou is the core region where the sample distribution is most concentrated. The terrain of this area is primarily plains with low hills, with an overall flat topography interspersed with rivers. The West Lake 西湖 district within the core region has the highest number of distributed samples, accounting for 35% within the entire range of Hangzhou 杭州.

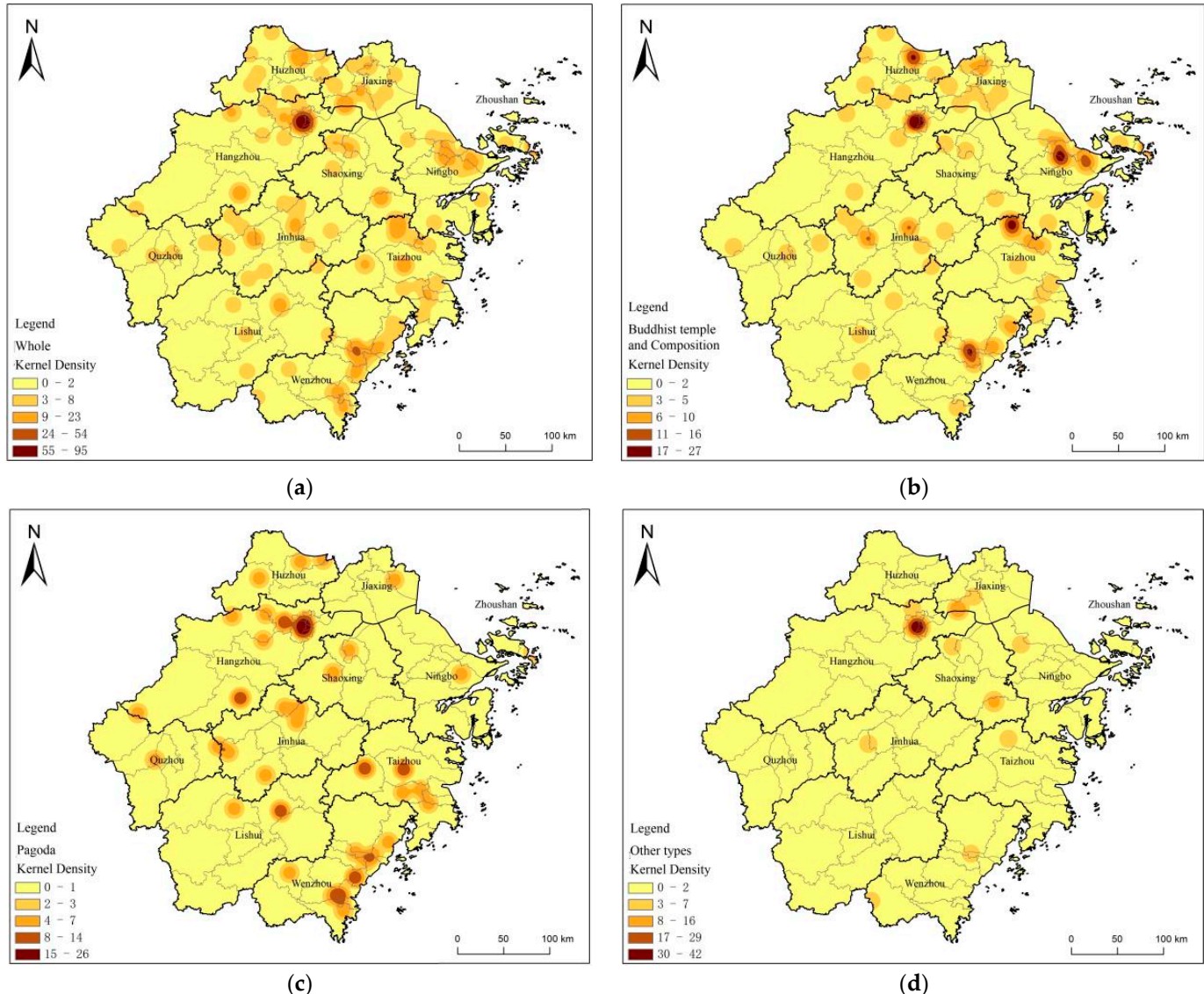

**Figure 2.** Kernel density of representative historical Buddhist architectural examples in Zhejiang Province: (**a**) kernel density of the entire area; (**b**) kernel density of Buddhist temples and constituent structures; (**c**) kernel density of pagodas; and (**d**) kernel density of Buddhist sutra pillars, statues, and grottos.

The state of uneven spatial distribution of representative historical Buddhist architecture in Zhejiang also distinctly manifests across various types of structures. The samples of Buddhist temples and constituent structures of temples exhibit a single primary core with multiple secondary cores in their distribution pattern, with the main core located in the central urban area of northeastern Hangzhou, where the output area-density per unit is the highest. There are four secondary cores, located in the northwestern part of Ningbo 宁波 and the central part of Huzhou 湖州, characterized by topographies primarily consisting of plains, low mountains and hills, and enriched with numerous lakes and a dense river network. The area around Mount Tiantai 天台 in the northern part of Taizhou 台州, the cradle of the Tiantai School, is mostly comprised of low mountains and hills. The central and southwestern parts of Wenzhou 温州 are composed of coastal plains and hills. The remaining samples, including pagodas, Buddhist sutra pillars, statues and grottos, all follow a distribution pattern with a single core in the central urban area of northeastern Hangzhou.

The spatial distribution of representative historical Buddhist architecture in the Zhejiang region exhibits discernible heterogeneity, resulting from the confluence of geograph-

ical, cultural, religious, conservation-related, and economic factors. Firstly, the geographical environment characterized by flat terrain and dense river networks is conducive to human activity, urban construction, and cultural development, thereby establishing these areas as the primary and secondary cores of Buddhist architectural examples. The central urban area in the northeastern part of Hangzhou is where the representative historical Buddhist architectural sites are best preserved and most densely distributed. This is attributed to Hangzhou progressively becoming the economic, cultural, and artistic center of the Zhejiang since the Tang Dynasty. Regions with economic prosperity are more likely to attract and sustain cultural, artistic, and religious activities. Consequently, the economic flourishing of Hangzhou has provided favorable conditions for the preservation and development of historical Buddhist architecture.

Additionally, Hangzhou was the capital of the Wuyue Kingdom (907–978) during the Five Dynasties and Ten Kingdoms period (907–979), as well as the capital of the Southern Song Dynasty (1127–1279). As a political epicenter, the support for Buddhism from the ruling class significantly stimulated the development of local Buddhist culture, providing a substantial material foundation for the construction and maintenance of Buddhist architecture. The urbanization of Hangzhou also led temples to assume social, cultural, and economic functions, making the city a primary core region for the distribution of large-scale temples. Furthermore, the religious background also influenced the significant aggregation of historical Buddhist architecture. Since the Song Dynasty, the Pure Land School 净土宗 gradually became an essential sect of Buddhism in Zhejiang, with Hangzhou serving as its activity center. The northern region of Taizhou 台州, particularly around Mount Tiantai 天台, is the birthplace of the Tiantai School 天台宗, making both locations of paramount religious significance and the epicenters of Buddhist activities in Zhejiang.

*3.2. The Temporal–Variation Characteristics of Representative Historical Buddhist Architecture in Zhejiang*

The representative historical Buddhist architectural sites in Zhejiang trace their origins back to the Three Kingdoms period (220–280) and continued to emerge until the Republican era (1912–1949), covering a span of approximately 1700 years (Table 2, Figure 3). In general, the spatiotemporal distribution reveals that earlier samples of representative historical Buddhist architecture in Zhejiang are predominantly located in the eastern and northern parts, with the western and southern parts showcasing more-recent developments. Among these architectural types, Buddhist temple samples emerged the earliest and exhibit the most sustained temporal continuity. The earliest samples of pagodas, grottos, and statues all originated during the Southern Dynasty, while the earliest samples of Buddhist sutra pillars appeared during the Tang Dynasty.

**Table 2.** Quantitative analysis of representative historical Buddhist architecture in Zhejiang across various historical periods.

| Construction Period | A.D. | Total | Buddhist Temples | Pagodas | Pillars, Statues, and Grottos |
|---|---|---|---|---|---|
| Three Kingdoms | 220–280 | 3 | 3 | | |
| Jin | 265–420 | 7 | 7 | | |
| Southern Dynasties | 420–589 | 19 | 14 | 2 | 3 |
| Sui | 581–618 | 2 | 1 | | 1 |
| Tang | 618–907 | 34 | 25 | 3 | 6 |
| Five Dynasties | 907–979 | 23 | 12 | 7 | 4 |
| Song | 960–1279 | 37 | 12 | 21 | 4 |
| Yuan | 1271–1368 | 10 | 2 | 5 | 3 |
| Ming | 1368–1644 | 17 | 7 | 9 | 1 |
| Qing | 1636–1912 | 5 | 3 | 2 | |
| Republic of China | 1912–1949 | 1 | 1 | | |

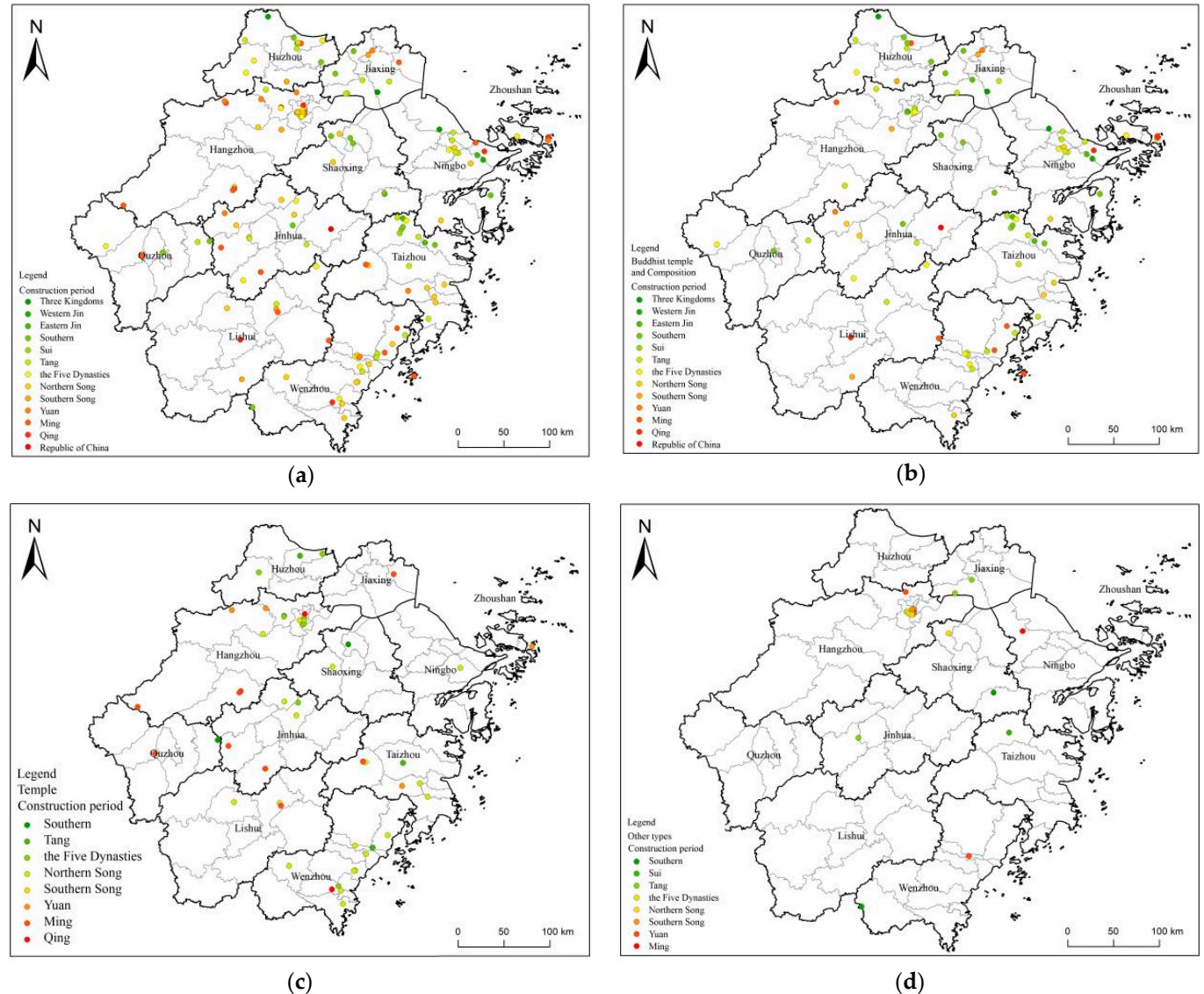

**Figure 3.** Distribution of the initial construction period of representative historical Buddhist architectural samples in Zhejiang Province: (**a**) distribution of the entire area; (**b**) distribution of Buddhist temples and constituent structures; (**c**) distribution of pagodas; and (**d**) distribution of Buddhist sutra pillars, statues, and grottos.

　　To more effectively study the temporal evolution characteristics of representative historical Buddhist architectural samples in Zhejiang, this study categorizes the samples into the following eight historical periods based on their initial construction years and quantities: Three Kingdoms to Eastern Jin (220–420), Southern Dynasties (420–581), Sui and Tang Dynasties (581–907), Five Dynasties (907–960), Song Dynasty (960–1297), Yuan Dynasty (1279–1368), Ming Dynasty (1368–1644), and Qing Dynasty to the Republican era (1644–1949). Utilizing the mean-center tool in ArcGIS, this study determined the distribution patterns of the sample's mean centers across the various historical phases (Figure 4). By observing the shifts in the location of the mean centers, it is apparent that there is a prominent trajectory moving from north to south, and it can be divided into four distinct directional phases.

　　In the first phase, the predominant trend in the movement of the mean center is a shift from north to south. During the period of the Three Kingdoms to the Eastern Jin, the mean center of historical Buddhist architectural samples is located in the northern part of Zhejiang. This center then began to shift southwestward in the Southern Dynasty and south-

eastward in the Sui and Tang Dynasties. In the second phase, the overall direction shift is from east to southwest. During the Five Dynasties, it moves to the northwest, and during the Song Dynasty, it migrates southward. In the third phase, the mean center shifts from south to north during the Yuan Dynasty. In the fourth phase, the mean center migrates from north to south, shifting southwestward during the Ming Dynasty and southeastward after the Qing Dynasty.

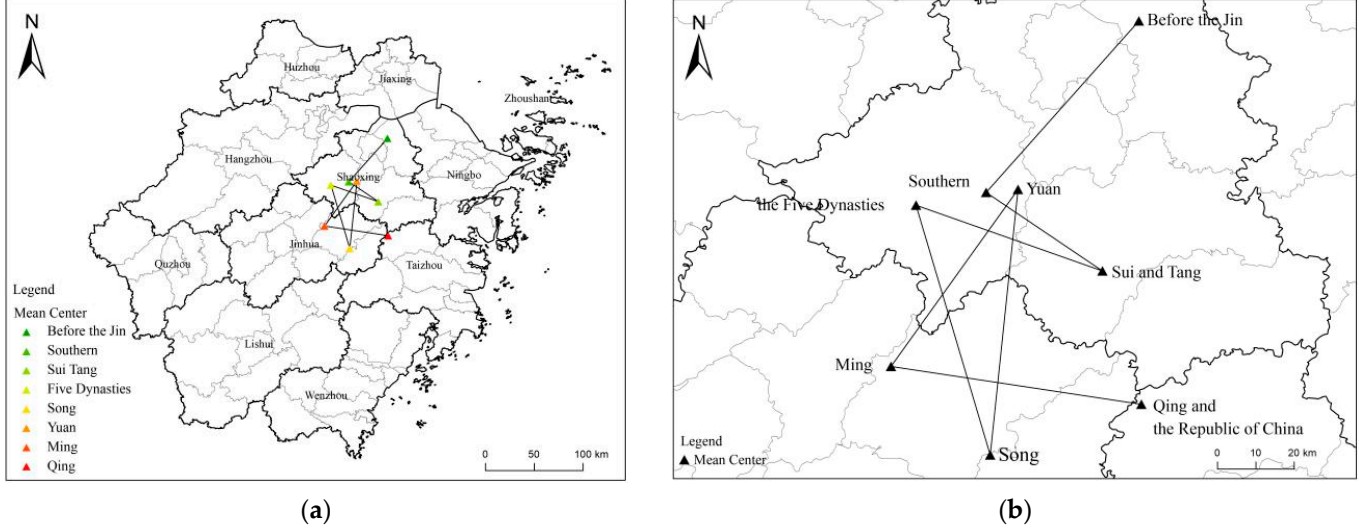

(**a**)　　　　　　　　　　　　　　　　　　　　　　(**b**)

**Figure 4.** Movement trend of the mean center of representative historical Buddhist architectural samples in Zhejiang: (**a**) movement trend of the mean center; (**b**) partial graph.

In view of the average nearest-neighbor index at the different periods (Table 3), the spatial distribution of samples from various historical periods exhibit three distinct characteristics. Initially, there is a progression from randomness to clustering, which subsequently transitions from a random to a dispersed distribution (Figure 5).

**Table 3.** Average nearest-neighbor index of representative historical Buddhist architectural samples in Zhejiang during different periods.

| Period | Observed Mean Distance/km | Expected Mean Distance/km | ANN | Z-Score | *p*-Value | Distribution Pattern |
|---|---|---|---|---|---|---|
| Total | 0.083997 | 0.129530 | 0.648475 | −8.479797 | 0 | Clustered |
| 220−420 | 0.299889 | 0.245532 | 1.221386 | 1.270583 | 0.203877 | Random |
| 420−581 | 0.253448 | 0.294557 | 0.860440 | −1.163771 | 0.244517 | Random |
| 581−907 | 0.170996 | 0.228147 | 0.749499 | −2.875353 | 0.004036 | Clustered |
| 907−960 | 0.334629 | 0.298270 | 1.121900 | 1.068668 | 0.285219 | Random |
| 960−1297 | 0.188505 | 0.208793 | 0.902831 | −1.130726 | 0.258170 | Random |
| 1279−1368 | 0.577736 | 0.400912 | 1.441055 | 2.668231 | 0.007625 | Dispersed |
| 1368−1644 | 0.456014 | 0.341167 | 1.365942 | 2.970161 | 0.002976 | Dispersed |
| 1644−1949 | 0.922661 | 0.505845 | 1.824001 | 3.861304 | 0.000113 | Dispersed |

During the period from the Three Kingdoms to the Southern Dynasties, the distribution trend of the samples appears to be random. Specifically, the samples from the Three Kingdoms to the Eastern Jin have an ANN of 1.22 (>1) and a z-score of 1.27, lying within the −1.65 to 1.65 range. Samples from the Southern Dynasties exhibit an ANN of 0.86 (<1) and a z-score of −1.16, also lying within the −1.65 to 1.65 range. Samples from the Sui and Tang dynasties commence a distinct clustering trend, with an ANN of 0.74 (<1) and a z-score of −2.87 (<−2.58). The Five Dynasties to the Song Dynasty revert to a random distribution, where samples from the Five Dynasties have an ANN of 1.21 (>1) and a z-score of 1.06, lying within the −1.65 to 1.65 range. Samples from the Song Dynasty hold

an ANN of 0.90 (<1) and a z-score of −1.13, also lying within the −1.65 to 1.65 range. From the Yuan Dynasty to the Republican era, there was a clear trend of dispersal. Samples from the Yuan Dynasty have an ANN of 1.44 (>1) and a z-score of 2.66 (>2.58). Samples from the Ming Dynasty have an ANN of 1.36 (>1) and a z-score of 2.97 (>2.58), and from the Qing Dynasty to the Republican era, the samples manifest an ANN of 1.82 (>1) and a z-score of 3.86 (>2.58).

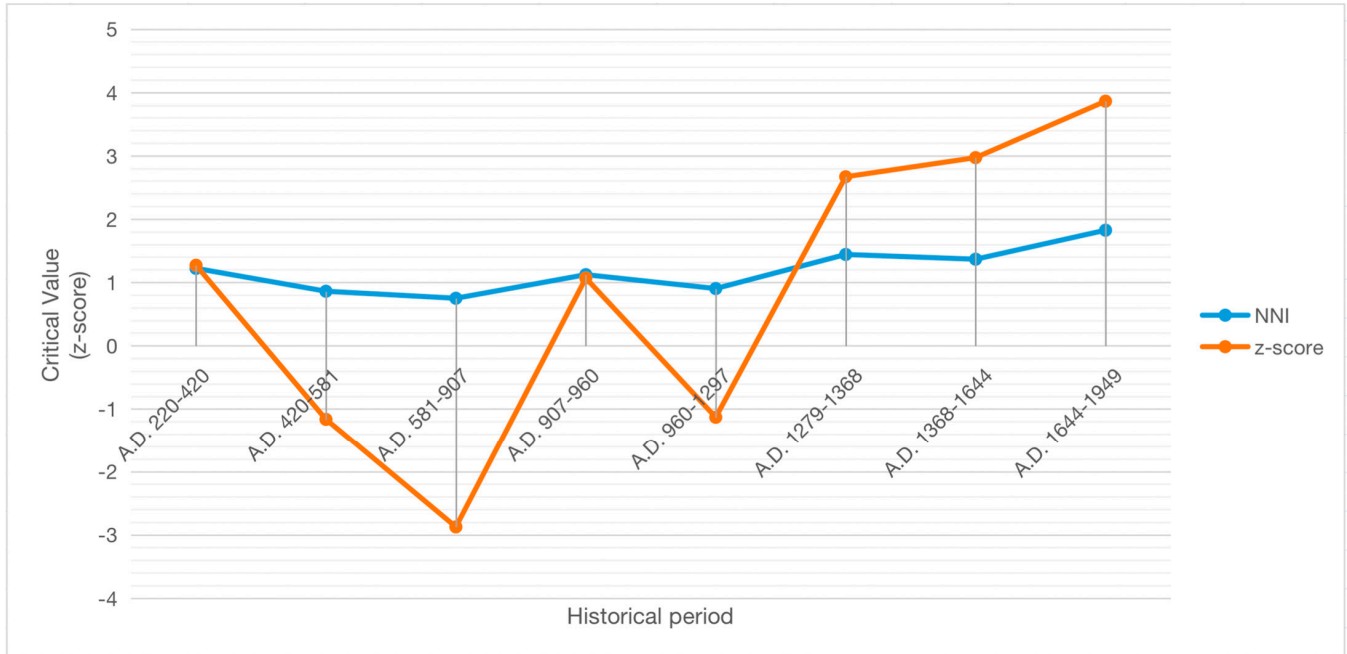

**Figure 5.** Average nearest-neighbor-index trend chart of representative historical Buddhist architectural samples in Zhejiang during different periods.

The trends in the quantity of representative historical Buddhist architectural samples in the Zhejiang and the spatial-distribution patterns of samples across different historical periods (Figures 6 and 7) can reveal the development trends of historical Buddhist architectural sites, reflecting the spatial extent and intensity of Buddhist activities.

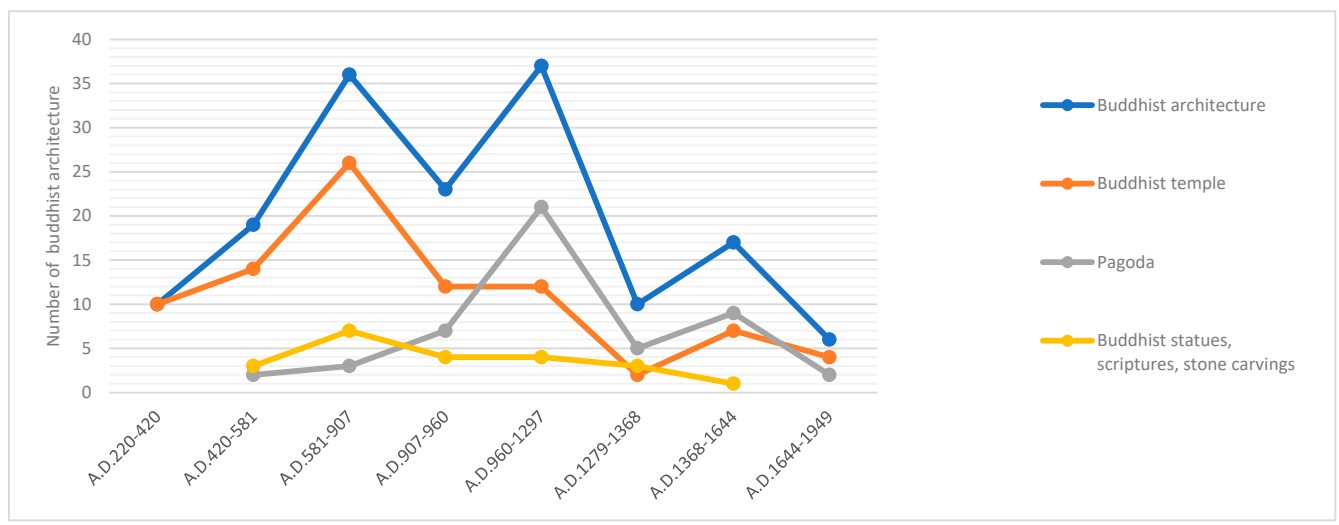

**Figure 6.** Spatial-distribution trend of representative historical Buddhist architecture of different types in different periods.

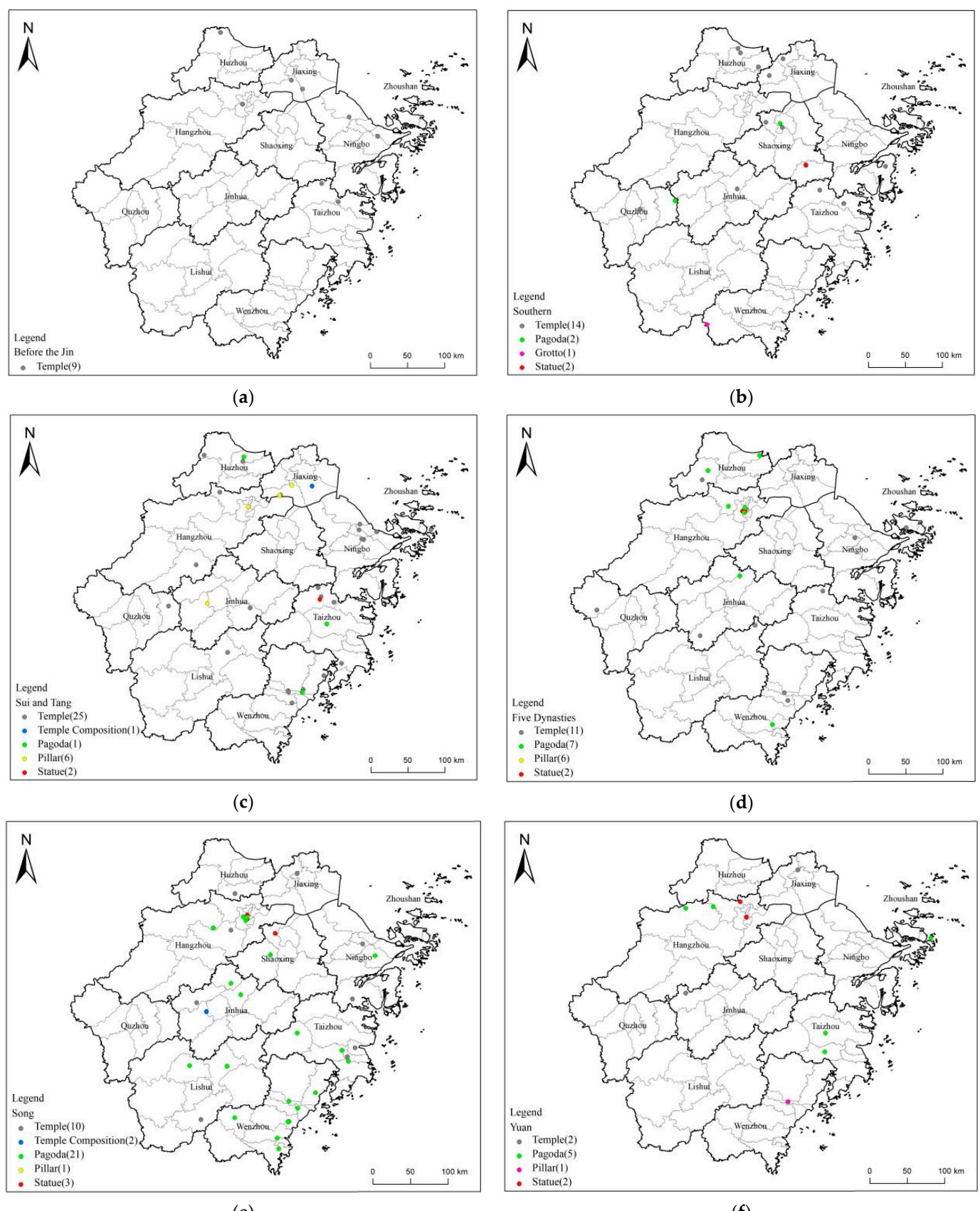

**Figure 7.** *Cont.*

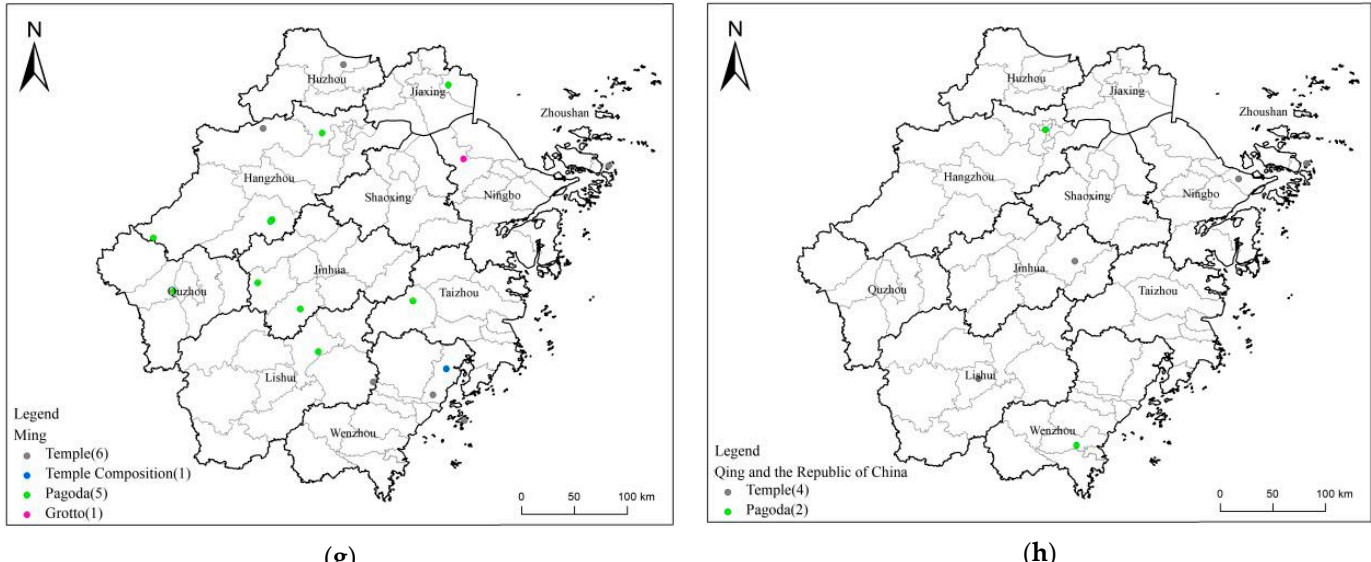

**Figure 7.** Distributions of new representative historical Buddhist architecture in Zhejiang in different periods: (**a**) distribution from the Three Kingdoms to Jin Dynasty; (**b**) distribution in the Southern Dynasty; (**c**) distribution in Sui and Tang Dynasties; (**d**) distribution in the Five Dynasties; (**e**) distribution in the Song Dynasty; (**f**) distribution in the Yuan Dynasty; (**g**) distribution in the Ming Dynasty; and (**h**) distribution from the Qing Dynasty to the Republican era.

The development trend of historical Buddhist architecture in Zhejiang demonstrates clear phases. The first phase, from the Three Kingdoms to the Eastern Jin Dynasty, marks the beginning of an upward trend in the number of samples and is also the initial period of Buddhist architecture's introduction to Zhejiang. Temples were the predominant form of historical Buddhist architecture, which are rather scattered in the northern part of Huzhou 湖州, the northeastern part of Hangzhou 杭州, the southern part of Jiaxing 嘉兴, and the northern part of Ningbo 宁波. The sporadic distribution of Buddhist architecture during this period is significantly influenced by the restrictive policies of the ruling class, which only permitted the construction of temples for monks from the western regions, while barring Han individuals from monkhood (Shi 1992, p. 352).

In the second phase, spanning from the Southern Dynasties to the Song Dynasty, the number of samples experienced two significant increases, marking a flourishing development stage for Buddhist architecture in Zhejiang, influenced by various factors including politics, culture, and technology. During the Southern Dynasty, the number of historical Buddhist architectural sites significantly increased, and the types of architectural sites became more diversified than before, witnessing the emergence of pagodas, grottos, and statues. The widespread dissemination of Buddhism facilitated the coexistence of new and old Buddhist architectural sites in the cosmopolitan areas of eastern and northern Zhejiang. The macroscopic distribution began to present a more sparse, networked array of temples, enhancing the connection between the Buddhist architectural sites.

During the Sui and Tang Dynasties, the number of historical Buddhist architectural samples reached its first peak. Temple samples became widespread across Zhejiang, and Buddhist sutra pillars emerged. Influenced by the sinicization of Buddhism, the entire province of Zhejiang began to showcase the initial scale of temple clusters. These clusters were characterized by a county-level network layout, exhibiting a denser distribution in the east and sparser in the west. During the Song Dynasty, the number of samples reached a second peak, especially in the number of pagoda samples. By then, centered around Hangzhou, a cohesive and interconnected system of Buddhist temples had been established throughout the Zhejiang Province.

In the third phase, from the Yuan Dynasty to the Republican era, there was a significant decline in the number of samples. This marked a downturn in the development

of historical Buddhist architectural sites; although, temples and pagodas experienced a slight increase.

## 4. Factors Influencing the Spatial-temporal Evolution of Representative Historical Buddhist Architectural Sites in Zhejiang

### 4.1. Natural Resource Endowment and Transportation Resource Utilization

Zhejiang is located in the southern part of China's Yangtze River Delta, experiencing a subtropical monsoon climate prone to frequent meteorological disasters. Its geographical coordinates range between 27° and 31° N latitude and 118° and 123° E longitude, respectively, encompassing a total area of 105,500 km². The southwest region is predominantly mountainous and hilly, featuring higher terrain, whereas the central region is characterized by hills and basins. The northeast region consists of a coastal accumulation plain, while the east is characterized by hilly and coastal plains, both presenting relatively lower terrain. Mountains account for 70.4% of the total area, with elevations not exceeding 2000 m. The province boasts numerous rivers and lakes, with well-developed water systems, and its eight major water systems span 1882.04 km in length. Rivers and lakes comprise 6.4% of the total area, and the coastline extends 6486 km. The regional geomorphic framework and environmental features remain essentially consistent between modern and historical times.

#### 4.1.1. The Relationship between the Spatial Distribution of Buddhist Architecture and the Terrain

Humans have a deep-rooted reliance on the natural environment, which has closely linked the scope of human activities to the spatial distribution of the Buddhist architecture. Elevation, as the principal attribute of topography, is a fundamental element of the natural environment. Different elevations present distinct conditions, including variations in climate, water resources, soil, and transportation. Consequently, elevation plays a significant role in influencing the distribution pattern of historical Buddhist architecture.

Overlaying historical Buddhist architectural samples with the DEM (Table 4, Figure 8), the samples located below 200 m in elevation characterized as plains constitute approximately 90% of the total, mainly representing urban and rural temples. As elevation increases, the number of samples gradually decreases. Samples located above 200 m in elevation in hilly and mountainous terrains make up around 10% of the total, predominantly consisting of mountain temples. It is evident that high-altitude areas are constrained by various factors. Inconvenient transportation impedes human mobility and the transportation of construction materials, thereby hindering temple construction. This limitation also affects the daily lives and sustenance of monks. Therefore, there are only six samples located above 1000 m in elevation.

**Table 4.** Spatial distribution of ground elevation of historical Buddhist architecture in Zhejiang Province.

| Altitude/m | 0–50 | 50–200 | 200–500 | 500–1000 | >1000 |
|---|---|---|---|---|---|
| Terrain type | Plain | Plain | Hills | Low mountain | Medium mountains |
| Buddhist architecture | 78 | 57 | 15 | 7 | 2 |
| Proportion | 0.49056604 | 0.35849057 | 0.09433962 | 0.04402516 | 0.03389831 |

#### 4.1.2. The Relationship between Spatial Distribution of Buddhist Architecture and Rivers

Water source, as a crucial material foundation for human production and living, are one of the main factors influencing spatial distribution and play a vital role in the selection of geographical locations for constructing temples. Utilizing ArcGIS, we drew maps depicting the distribution of representative historical Buddhist architectural samples and river buffer zones in Zhejiang, conducting a buffer analysis of the rivers within 0–1 km, 1–3 km, 3–5 km, 5–10 km, and over 10 km (Table 5, Figure 9). The analysis shows that the distribution pattern of the samples along the Yangtze River and its tributaries tends

to form belt-like temple clusters. The distribution of the samples varies significant across different river buffer ranges. 61% of the samples are located within 5 km of a river, and the number of samples diminishes as the distance from the river increases. It is clear from these findings that historical Buddhist architectural sites often favored locations near water, reflecting their affinity for water proximity.

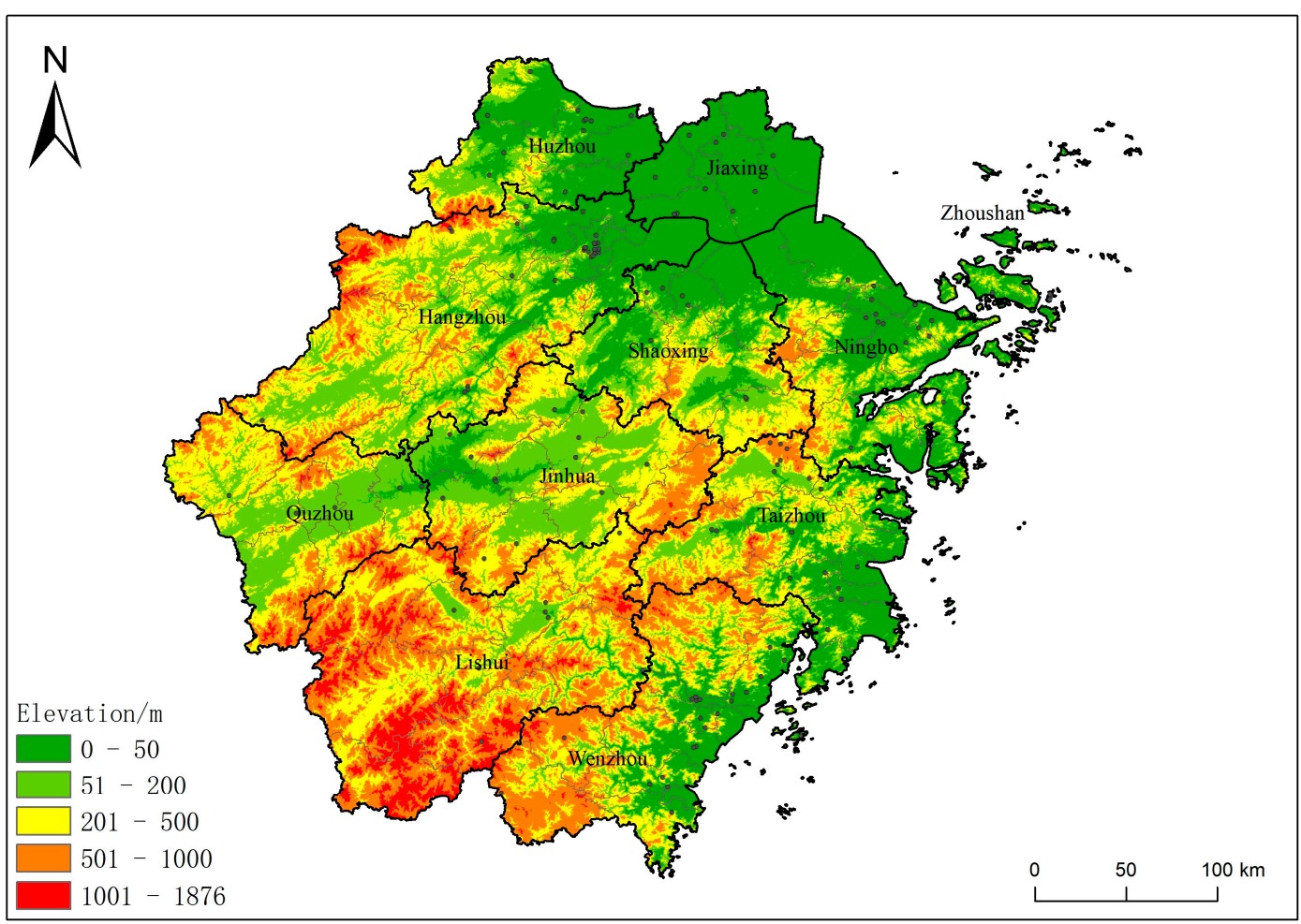

**Figure 8.** Spatial distribution of ground elevation of historical Buddhist architecture in Zhejiang Province.

**Table 5.** Spatial distribution of river buffer ranges of existing Buddhist architecture and representative historical Buddhist architecture in Zhejiang Province.

| Buffer Range/km | 0–1 | 1–3 | 3–5 | 5–10 | >10 |
|---|---|---|---|---|---|
| Buddhist architecture | 50 | 28 | 20 | 17 | 44 |
| Proportion | 0.314465409 | 0.176100629 | 0.125786164 | 0.106918239 | 0.27672956 |

### 4.1.3. The Relationship between the Spatial Distribution of Historical Buddhist Architecture and Transportation

Transportation, serving as a vital conduit for economic development and cultural exchange, can significantly broaden and deepen human economic and cultural activities, enrich both spiritual and material civilization, and increase the depth and breadth of the exchanges. Transportation routes act as crucial avenues of penetration, for example, ancient post roads provided a readily available channel for religious dissemination (Cao and Xu 2005, pp. 25–31). The convenience of transportation influences, to some extent, the spatial distribution of historical Buddhist architectural sites. Post roads[8] within Zhejiang

were established during the Tang and Song Dynasties, and remained in use until the Republican era. These roads had a certain aggregating effect on people's modes of activity and were closely related to the construction of Buddhist architectural sites. The layout of modern roads essentially follows the pattern set by the ancient post roads (Luo 2022, pp. 77–84, 112).

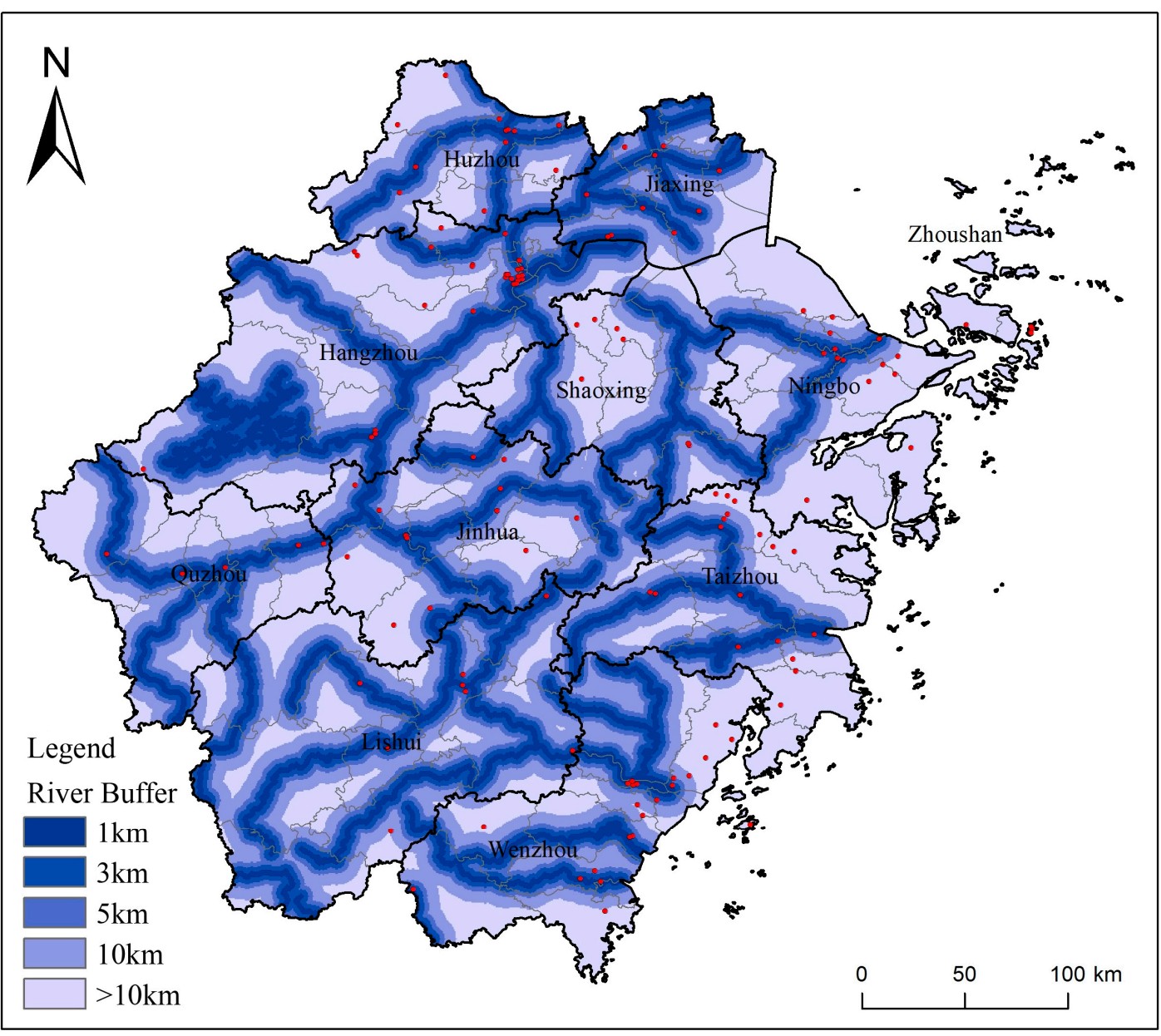

**Figure 9.** Spatial distribution of river buffer of representative historical Buddhist architecture in Zhejiang Province.

In this study, main roads[9] within Zhejiang were used to create road buffer zones and map the distribution of representative historical Buddhist architectural samples. The results reveal that samples within an 8 km buffer range are distributed along the main roads of Zhejiang, accounting for 88.4% of the total in the province (Table 6, Figure 10). Convenient transportation enhances the accessibility and use of Buddhist sites, leading to a distribution with a pronounced orientation towards transportation.

**Table 6.** Spatial distribution of road buffers of representative historical Buddhist architecture located in Zhejiang Province.

| Buffer Range/km | 0–1 | 1–2 | 2–4 | 4–8 | >8 |
|---|---|---|---|---|---|
| Buddhist architecture | 54 | 20 | 28 | 41 | 16 |
| Proportion | 0.33962264 | 0.12578616 | 0.17610063 | 0.25786164 | 0.10062893 |

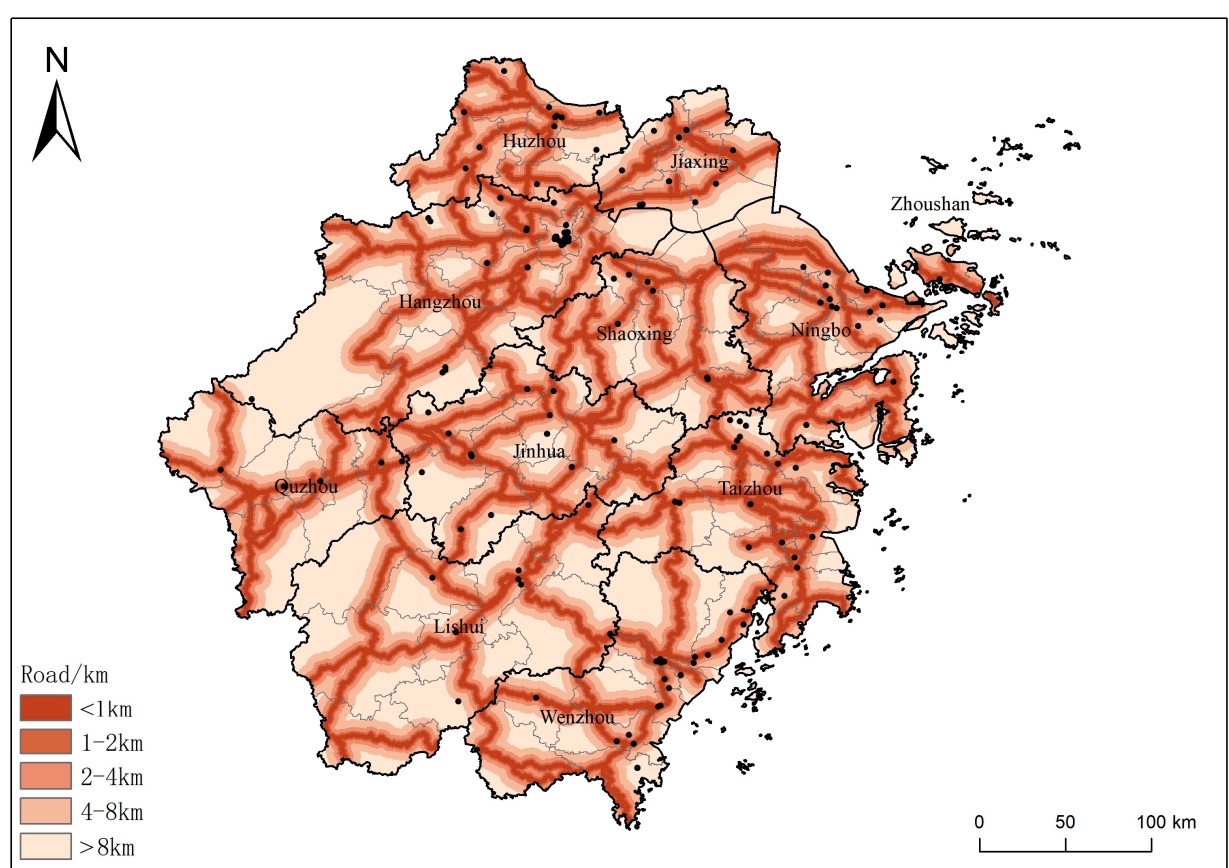

**Figure 10.** Spatial distribution of road buffers of representative historical Buddhist architecture located in Zhejiang.

*4.2. Development and Evolution of Buddhist Architecture under the Influence of the Buddhism's Secularization Trend*

As Buddhism tends towards secularization, various factors such as culture, politics, economy, and technology have significant impacts on Buddhist architecture. By examining the three phases of spatial-distribution changes of historical Buddhist architecture in Zhejiang, we can delve deeply into the influence of secular factors on the development and evolution of Buddhist architecture.

4.2.1. The Initial Introduction and Integration Phase of Buddhist Architecture

In the Han Dynasty, the dominant religious and cultural foundation of Chinese society was primarily grounded in Confucianism, intertwined with concepts of yin-yang 阴阳, the five elements, and the synchronicity of heaven and human beings, with Confucianism bearing the most crucial role in moral education within the society. The introduction of Buddhist culture to Zhejiang disrupted this indigenous religious equilibrium, gradually leading to an intricate interplay among Confucianism, Buddhism, and Daoism. This shift led to the emergence of syncretic worship within local religious structures, where deities from the three religions were revered simultaneously. Confucianism, distinguished by

its dual facets of politicization and religiosity, channeled and integrated Buddhist culture through ritualistic education, driving the sinicization trend of Buddhist culture.

During this period, the growth in numbers and spatial distribution of Buddhist historical architecture in Zhejiang was predominantly influenced by the intersection of Confucian ideological culture, feudal hierarchy, and construction technology. This manifested in the adaptation of Buddhist architecture to the hierarchical system under Confucian rites and norms. From the Three Kingdoms to the Eastern Jin period, societal fragmentation due to warfare and the resultant instability in people's lives provided a conducive environment for the propagation of Buddhism. Buddhism experienced rapid dissemination and development amidst conflicts and integration with the indigenous Confucian and Daoist cultural philosophies of China. Observing the elevation distribution of Buddhist historical architectural samples from this period (Figure 11), the primary forms of samples were urban temples located in plain terrains and mountain temples situated in hilly landscapes below 1000 m of elevation.

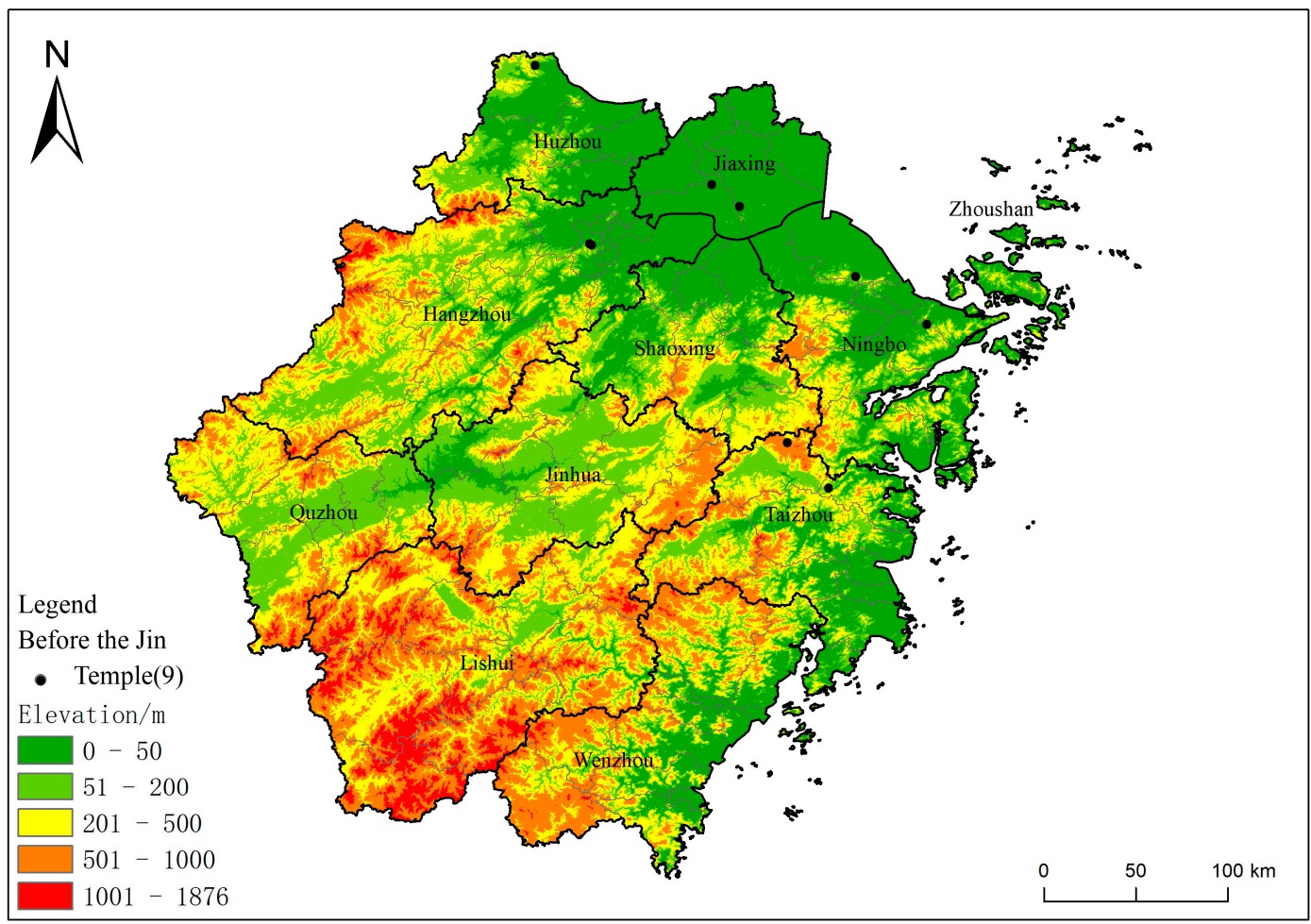

**Figure 11.** The elevation distribution of Buddhist historical architecture samples from the Three Kingdoms to the Eastern Jin period.

Influenced by local Chinese culture, the Western Region monks having resided in China for an extended period, and have advocated the convergence of the three religions: Confucianism, Buddhism, and Daoism. They translated Buddhist scriptures using the terminologies and theoretical concepts of Confucianism and Daoism, such as the concept of original non–being in Daoism and the ethical views of benevolence, righteousness, filial piety, and kinship in Confucianism (Hong 2002, pp. 81–93). This integration of Buddhist culture fostered a value system grounded in practical living, prompting the ruling class to

repurpose existing buildings in their residences within Buddhist temples, gradually establishing a trend. Consequently, many early urban and suburban Buddhist temples evolved from existing residential structures, reflecting the form of traditional Chinese dwellings, abandoning the original residential functions and assigning new Buddhist functions. For instance, the Puji 普济 Temple built in Cixi 慈溪, Zhejiang, during the Eastern Wu period (229–280), was converted by Kan Ze 阚泽, the Senior Tutor to the Crown Prince of Wu, from a book hall in his residence (Zan 1987, p. 639). The litterateur Xu Xun 许询, of the Eastern Jin Dynasty (317–420), converted his old houses located in Shanyin[10] 山阴 to Zhihuan 祇洹 Temples, and his new residence in Yongxing[11] 永兴 to a Chonghua 崇化 Temple (Xu 1986, p. 10).

Urban Buddhist temples blurred the distinction between the sacred and the secular, by embracing secular architectural forms as venues for Buddhist activities to enhance public identification with Buddhism, becoming a significant factor in the localization and secularization of Buddhist architecture. In contrast, mountain temples, influenced by Buddhist doctrines advocating detachment from the secular world in pursuit of inner peace, clearly separated the sacred from the secular.

With the rise of Mahayana Buddhism during the Eastern Jin Dynasty, monks started to establish settlements, leading to a significant increase in urban temples. By this period, local construction technology had reached advanced craftsmanship levels, with techniques such as timber framing, stone carving, and mural painting being widely employed in the design and construction of temples. Given the need for the architectural planning of temples to cater to the translation and interpretation of scriptures, discourse, and interaction inherent in Buddhist education, educational institutions and lecture temples emerged, these establishments prioritized teachings and practices for monks, intellectuals, and the general populace. These Buddhist structures predominantly feature pagodas and Buddha halls as the main bodies, supplemented with other auxiliary buildings. The lecture halls within serve as spaces for monks to interpret scriptures and learn, and do not venerate statues of Buddha. The main structures are situated behind the pagoda, and arranged along the central axis according to traditional architectural layouts, forming a solemn arrangement dominated by multiple courtyards, flanked by constituent structures on both sides. These constituent structures, usually the monasteries, warehouses, and kitchens for monks, are established surrounding the pagoda and Buddha hall.

Simultaneously, mountain temples, commonly known as Jing She 精舍 (Fang 1974, p. 231), were typically constructed by monks who cleared forests and carved mountains. Compared to urban Buddhist temples, mountain temples exhibit relatively modest architectural styles and more flexible layouts, adopting the forms of mountain residences (Fu 2001, p. 157). They cleverly integrate with the terrains and landscapes, deconstructing the architectural layout into more versatile arrangements, typically manifesting as small-scale grassroots and thatched cottages and sheds without the establishment of Buddhist pagodas.

The earliest mountain Buddhist temple in the Zhejiang appeared during the Yongkang 永康 (300–301) period of the Western Jin Dynasty. The monk Yixing 义兴 traveled to the East Valley of Taibai 太白 Mountain in Yinzhou 鄞州, Ningbo 宁波, constructing Jing She amongst the mountains, which is the predecessor of Tiantong 天童 Temple (Lu 2015, p. 261). During the Eastern Jin Dynasty, Buddhism witnessed a downturn, merging with metaphysical studies. Many monks and laymen began to retreat to the mountains, and the number of Buddhist temples, which combined the functions of seclusion and preaching, gradually increased. For example, Zhong Fangguang 中方广 Temple, located in Tiantai 天台 Mountain in Zhejiang, was initially built during the Xingning 兴宁 period (363–365) during the Eastern Jin Dynasty. Additionally, in the first year of the Xianhe period (326–334) in the Eastern Jin Dynasty, Hui Li 慧理[12], a monk from Western India, established the Lingyin 灵隐 Temple and Lingjiu 灵鹫 Temple beneath the Feilai 飞来 Peak (Tian 1958, p. 125).

4.2.2. The Flourishing Development of Buddhist Architecture under the Influence of Diverse Societal Factors

Starting from the Southern Dynasties, the number of historical Buddhist architectural samples notably increased, with Shaoxing and Hangzhou successively becoming regional Buddhist centers, with a dense concentration of Buddhist temples in the Jiangnan area. The development of Buddhist architecture entered a stage of flourishing growth. The most prominent influential factor during this period was the formation of a hierarchical system in Buddhist temples, characterized by the large-scale landscaping, and privatization of Buddhist temples. Concurrently, the increase in the number of samples, the diversification of types, and the formation of temple networks also played a significant role in promoting the syncretism of the three teachings and the secularization of Buddhism.

The establishment of the monastic official system[13] effectively led to a merger of politics and religion, affirming the position of monks at religious, social, and political levels, granting them certain privileges and status. This system reinforced the importance and hierarchical order of religious architecture, both materially and symbolically. The ruling class sustains the material foundation for the propagation of Buddhism through monetary offerings, construction of temples, and land donations. Especially after a large number of Buddhist temples acquired land, it elevated the proportion of the temple economy in the overall social economy, providing financial resources for the expansion of Buddhist activities and the construction of Buddhist temples. Simultaneously, the stringent hierarchical system of secular architecture in the Zhejiang region has explicit provisions for physical spaces representing power, status, and wealth, providing a clear framework for the hierarchical system of Buddhist architecture. Therefore, with the combination of religious and secular hierarchical concepts, the style and spatial layout of Buddhist architecture have become more defined and standardized, rendering it a product of the stringent hierarchical system marked by distinctions between superior and inferior, and differences between the noble and the humble.

Additionally, with the dissemination of the Lotus Sutra, Buddhism has evolved towards secular idolatry. Offering to Buddhist statues has become a widely accepted mode of worship among the mainstream society in Zhejiang. For a place of worship, the Buddhist temple serves as a sanctum for enshrining representations of the Buddha, embodying the devout reverence and deep respect people have for the Buddha and Buddhist teachings. It is the core of Buddha worship. This period saw the emergence of grottoes and statues, such as the giant stone Maitreya Buddha carved in Xinchang 新昌, Shaoxing 绍兴, and initiated the grotto statues of the Thousand Buddha Academy during the Yongming (483–493) of the Southern Qi Dynasty (479–502).

Under the influence of the imperialization of Buddha statues, the ruling class began to cast large-scale Buddha statues and establish halls to house them. It was common to establish multiple Buddha halls within a single monastery, and halls specifically dedicated to venerating seven Buddha statues also emerged (Xu 1986, p. 411), as seen in the numerous large and small halls in the Tongtai 同泰 Temple established during the Southern Liang Dynasty (502–557) (Xu 1986, p. 681). The establishment of multiple Buddhist halls promoted the large-scale development of the layout of Buddhist architecture in Zhejiang, resulting in the emergence of subsidiary courtyards located around the main buildings, featuring relatively flexible layouts.

During the Sui and Tang dynasties, the economy of Buddhist temples further developed, providing the economic foundation for the establishment of a large number of Buddhist sects[14]. The number of historical Buddhist architectural samples also significantly grew, reaching its first peak in history, with the emergence of four concentrated regional distributions of Buddhist temple settlements (Figure 12). Ningbo 宁波, Wenzhou 温州, and Taizhou 台州 are the main ports of the Maritime Silk Road. Influenced by the route of Buddhism spreading northward along the Maritime Silk Road, these areas were the first to form spatially concentrated settlements of small Buddhist temples. In addition to these, Huzhou 湖州 and Jiaxing 嘉兴 also, with the widening and regulation of the Jiangnan

Canal, connected northward to Zhenjiang 镇江 in Jiangsu 江苏, southward to Hangzhou 杭州, and eastward to Shaoxing 绍兴 and Ningbo. With the opening of water routes, and the implementation of the policy promoting the construction of Buddhist temples in areas with monk activities (Dao 2014, p. 549), they gradually formed larger networked temple settlements.

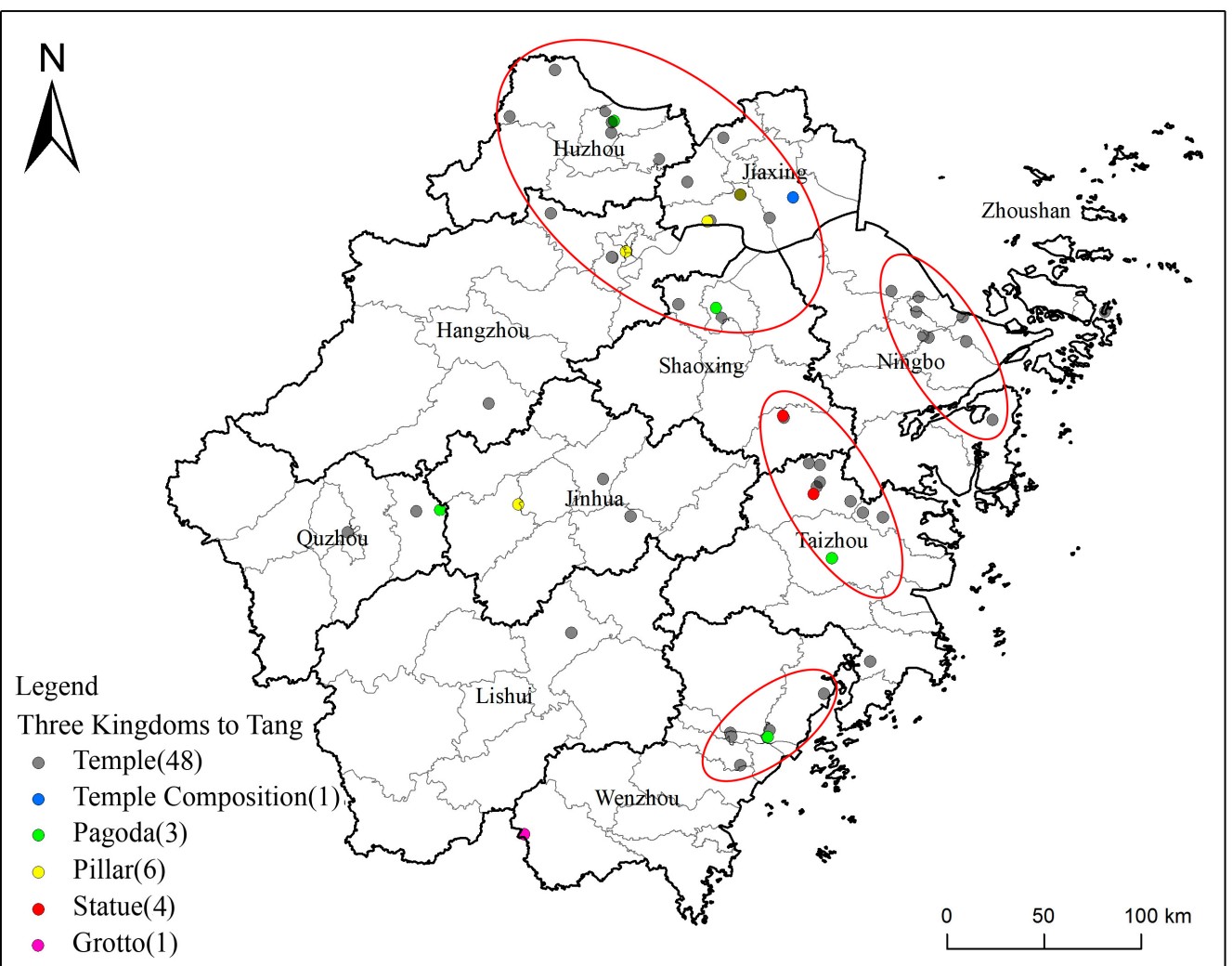

**Figure 12.** Network of Buddhist historical architectural sample settlements during the Tang Dynasty.

During the Tang Dynasty, Buddhist activities and their architectural construction were integrated into the national economic and political systems, becoming a stable and orderly presence within the spiritual and cultural realms of Zhejiang society. The construction of Buddhist architecture was politically contingent upon the secular society's power hierarchy and economically dependent on the offerings of the followers. The dominant power of the ruling class and patrons became one of the significant factors leading to the secularization of Buddhist architecture.

Following the policy of establishing Buddhist temples according to administrative regions of prefectures and counties (Dao 2014, p. 940), and the implementation of the three principles 三纲 system[15] for temples, various regions enacted a pre-application system within a prescribed quota (Liu 1975, p. 1831), leading to a distinction between official and folk temples in nature (Fu 2001, p. 472), and the gradation of temples based on scale, importance, and function. Among them, imperial temples built by the emperor or royal family members held the highest status, followed by large official temples built with granted quotas and funded by local–government officials and eminent monks. Representative exam-

ples of Buddhist historical architecture mostly belong to these two categories. Conversely, privately operated Buddhist halls within counties, aranya 兰若[16] constructed by monksl (Li 2021, p. 37), and mountain retreats or rural Buddhist halls donated by the general populace are all considered unofficial, medium– to small-scale folk temples and hold the lowest rank.

Driven by both power and utility, the official temples transformed from single architectural entities into complex structures consisting of a central courtyard and multiple subsidiary courtyards, evolving in directions that are more grandiose, intricate, and stable. Advancements in wooden construction, including pillar and beam connections, sophisticated tile-making, ornamentation techniques, along with mural painting and colored sculpture crafts, have provided technical support for this transformation. The enlargement of Buddhist architecture has also fostered the conduct of solemn and grand scripture lectures, bolstering the spread of Buddhist education and heightening its societal influence. This evolution further cemented the secularization of Buddhist venues. For instance, monks interpreted stories from the scriptures into secular tales relatable to the daily lives of the common people and narrated them to ordinary urban dwellers within these extensive Buddhist spaces.

Influenced by topography, landscape, and societal aesthetic preferences, mountain temples, embodying natural harmony and transcendent serenity, emerged as the most distinctive form of Buddhist architecture during the Tang Dynasty. Consequently, the quantity of historical Buddhist architectural samples located in mountainous and hilly terrains displayed a growing trend, with the mountainous hills of eastern Zhejiang becoming a concentrated area for such temples. Mountain temples are typically constructed alongside high mountain ranges and cliffs that cascades and unfolds according to the mountain's terrain, surrounded by woods and winding streams. The serene isolation created in the deep mountains satisfied the monks and scholars' desires for secluded living and dedicated cultivation, giving rise to two typical modes of Buddhist education. One mode of Buddhist education primarily involves chanting scriptures and cultivating the mind within these mountain temples, as exemplified by monk Shi Hanshan 释寒山, who spent years in seclusion at Cuiping 翠屏 Mountain in Taizhou 台州, composing poems and writing (Fu 2009, p. 306). Another mode inherits the traditional educational approach from the Jin and Southern Dynasties, where disciples were gathered for private lectures within mountain temples. For example, monk Huiyin 慧因 lectured in mountain temples for 30 years and taught over 500 disciples (Dao 2014, p. 431).

During the Song Dynasty, the hierarchical system of temples continued to adhere to Confucian ceremonial norms. The merging of Buddhist cultures from both the north and south, coupled with a growing need for spaces dedicated to Buddhist activities and the burgeoning temple economy, collectively propelled a significant increase in the number of historical Buddhist architectural sites, signifying the second historical peak. The spatial distribution of the samples has already exhibited a core-edge characteristic. During this period, an integrative large-scale network of Buddhist temples emerged, radiating outward from the core of Hangzhou (Figure 13). In the context of the highly secularized development of Buddhism, the ruling class strengthened the restrictions on the construction of Buddhist architecture. This led to a pronounced emphasis on aesthetics and landscaping in Buddhist architectural designs. Additionally, the flourishing temple economy exerted a profound influence on the functionality and stylistic transformations of Buddhist architecture.

First, the influence of Confucian ceremonialism led to the Confucianization and formalization of Buddhist architecture. During the Song Dynasty, Neo-Confucianism, representing the central ideological stance of the government, gained significant momentum. In order to accommodate the need for strengthened centralization in feudal society, the Buddhist organization during the Song Dynasty became subservient to the ruling class. The ruling stratum continually augmented restrictions on the establishment of Buddhist constructions, leading to the formation of the hierarchical system of official temples, known as

"the Five Mountains and Ten Monasteries 五山十刹"[17] (Lang 2009, p. 55). Notably, most of these high-level temples were located in the Zhejiang. This facilitated the emergence of a "core-edge" distribution characteristic of temple networks of major temple clusters in Zhejiang, with Hangzhou, the administrative and cultural center, serving as the core. The philosophy of synthesizing Confucianism, Daoism, and Buddhism gradually, rooted in Confucian thought, became mainstream in the development of native Chinese philosophical thoughts.

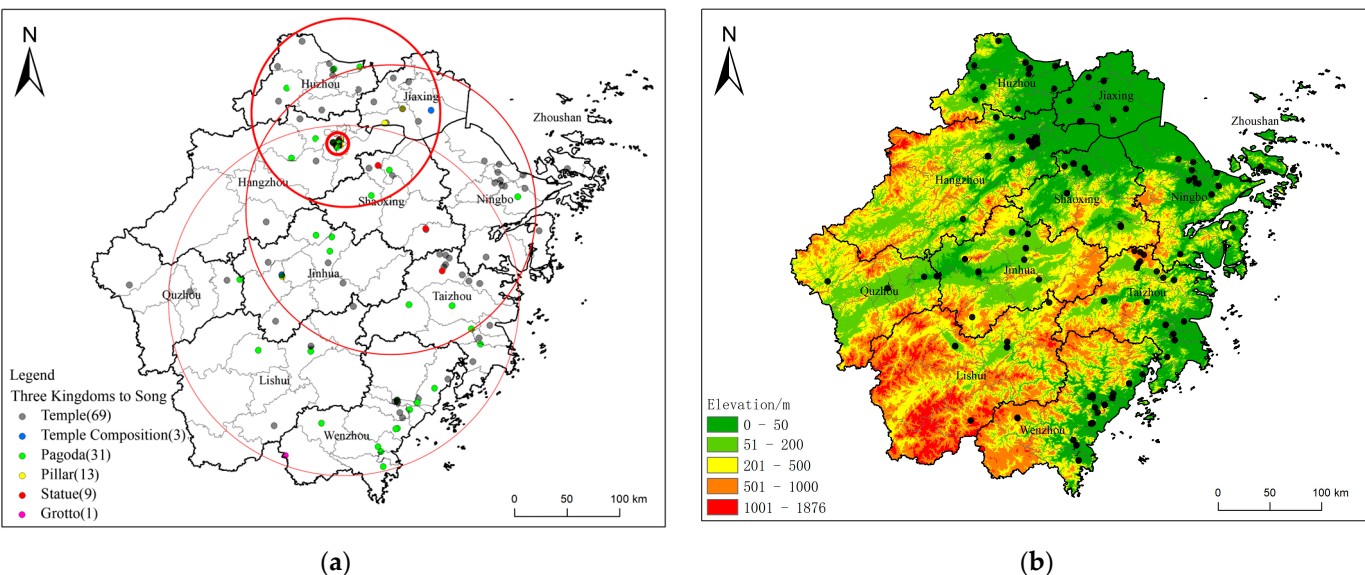

(**a**)          (**b**)

**Figure 13.** Spatial distribution of Buddhist historical buildings in Song Dynasty: (**a**) network of Buddhist historical architectural sample settlements; (**b**) the elevation distribution of Buddhist historical architectural samples.

Monks began to advocate for propagating Buddhism with Confucianism as the foundation (Hong 2002, pp. 81–93), emphasizing the theoretical integration of Buddhism with Confucianism and Daoism. Consequently, the *Ying Zao Fa Shi* 营造法式[18] (Pan and He 2017, p. 137), a construction regulation explicitly based on Confucian ceremonials, incorporated Buddhist architecture into the official construction of the hierarchical system, rigorously standardizing materials and labor and highlighting the formal order in the construction of official Buddhist temples. Apart from unique Buddhist structures, such as shrines and scripture cabinets, most structures adhered to the design standards of secular official buildings in terms of appearance, scale, materials, component forms, and fabrication methods, promoting standardized design and modular construction. The lotus and Baoxiang 宝相 patterns, commonly seen in Buddhist art, were also applied to beam-columns, lintels, rafters, columns, and arches of secular buildings, facilitating the secularization of Buddhist art.

Secondly, the syncretism of Confucianism, Daoism, and Buddhism has influenced the landscaping and incorporation of natural scenery in Buddhist architecture. Preferred locations for mountain temples in the central and southern parts of Zhejiang steadily gravitated towards higher elevations, with many appearing in regions above 500 m in elevation, some even exceeding 1000 m. Large mountain temples situated in tranquil and forested landscapes focus more on artificial shaping of the surrounding environment, displaying a layout characterized by a blend of solemnity and freedom.

The main structures positioned along the central axis, influenced by Confucian thought, manifest a distinct hierarchical order. In contrast, the ancillary buildings are designed to be in harmony with their surroundings, influenced by the Daoist principle of conforming to nature (Wang 2009, p. 150), such as the 20 miles of ancient pines lining both sides of the pathway in front of Tiantong 天童 Temple and streams leading to Lingyin

Temple. During the Song Dynasty, the construction of Buddhist temples placed significant emphasis on landscape and garden creation. This approach was, on the one hand, influenced by the literati class's aesthetic appreciation for natural landscapes and their spiritual pursuit of utopian refuges. On the other hand, it was closely connected to China's intrinsic cultural elements, especially the Daoist pursuit of health in the natural environment[19] and the Confucian culture spirits of transcending and engaging with the world[20].

The teachings of Buddhism champion the release from earthly ties and the pursuit of inner tranquility and wisdom, resonate with the Confucian ascetic elements, displaying symbiosis and complementarity in personal cultivation, moral values, and social harmony. Therefore, in the exchange and interaction between Confucianism and Buddhist culture, Confucian scholars have attempted to integrate the two, cultivating an integrated philosophical system of Confucianism, Buddhism, and Daoism, by utilizing mountain temples as a vehicle, aiming to find a balance between personal cultivation and social development, further propelling the secularization of Buddhist architecture. Mountain temples also incorporated elements of Daoism and Confucianism within their architecture and practices, fostering a convergence among the three religions and providing a platform for their cultural exchange.

Thirdly, the highly developed temple economy influenced the privatization of Buddhist architecture and the secularization of Buddhist art. Due to the urban economic development in the Song Dynasty and the gradual liberation of individuals from dependent relationships, the temple economy also followed the development trend of secular society, shifting from a feudal lord–based economy to a landlord economy (Chen 2019, pp. 157–63), increasingly aligning with secular economics in terms of management. During this period, the turmoil in the north caused a massive migration of people to the south (Li 1988, p. 1422), resulting in a significant population increase in Zhejiang. This led to a substantial increase in the number of Buddhist followers, jointly stimulating the rapid development of the temple economy.

On the one hand, this led Buddhist temples in Zhejiang to take advantage of their own economic strength and actively participate in local charitable activities, such as building bridges, roads, water conservation projects, elderly care, and famine relief (Huang 1989, p. 435). At the same time, the demand from the followers for Buddhist activity spaces not only increased the number of temples but also elevated their economic reliance with local believers, promoting the level of temple privatization. Following the establishment of the sub-temples system (Tian 1958, p. 117), where private temples are affiliated with official temples, shifts began to be seen in architectural layouts, resulting in the emergence of the "seven halls saṃghārāma 七堂伽蓝"[21] style (Li and Bai 2011, p. 134) in Chan Buddhist architecture. Examples of this can be seen in Lingyin 灵隐 Temple in Hangzhou, Tiantong 天童 Temple in Ningbo, and Wannian 万年 Temple in Tiantai 天台 Mountain. The burgeoning demands posed by Buddhist activities, ranging from temple edification and Buddha statuary to intricate architectural ornamentation, fueled the growth of secular artisanal industries and invigorated the commodity marketplace. This development led to the emergence of craftsmen and artisans who specialized in disciplines such as scripture transcription, statuary, and portraiture, catalyzing a wave of specialization within vernacular handicraft sectors. Concurrently, the overarching influence of Buddhist culture resonated deeply within the stylistic paradigms of fields like architecture, sculpting, furniture craftsmanship, embroidery, and painting.

On the other hand, the reliance of Song Dynasty Buddhist architecture on financial support from lay believers also propelled the popularization of Buddhist architectural art and the incorporation of entertainment functions within the temples. A significant example of the secularization of Buddhist architectural art, born from the fusion of folk beliefs, Pure Land thoughts, and Chan philosophy, is the statue of the Budai 布袋 Maitreya, located in niche number 68 at the Feilai Peak in Hangzhou. The statue depicts the Budai Maitreya in a relaxed and approachable manner, characterized by a slightly wrinkled forehead, a fat belly, a bare chest, and a wide smile. The figure's seated posture is relaxed and unbur-

dened by formalities, with legs spread open and knees slightly bent. His left hand holds a string of prayer beads, while his substantial right hand rests on a cloth sack. Given that the teachings of Budai Maitreya are rooted in worldly joy and steered by benevolent intentions, intended to guide the followers' worldview, the portrayal of the Budai Maitreya is created to be more approachable, resembling a more amiable monk in real life. This stands in sharp contrast to the stylized Buddhist statues before the Song Dynasty, which were either dignified and elegant or solemn and serene, symbolizing the profound integration of Buddhist imagery into everyday life during a period marked by a prevailing trend of secularization of Chinese culture.

In addition, the migration of monks and believers from the north fostered a further amalgamation of Buddhist culture and religious rites between the north and the south. Influenced by Tibetan Buddhist art, the architectural style of northern Buddhist architecture steered urban Buddhist architectural style in the Zhejiang towards a grander and more solemn aesthetic. As the capital of the Southern Song Dynasty (1127–1279), Hangzhou wielded significant influence over the evolution of Buddhist architecture in Zhejiang. The ruling class, on one hand, significantly increased the number of Buddhist temples, on the other hand, appropriated some of them as imperial gardens and private retreats, thereby endowing Buddhist architecture with secular functions of leisure and recreation.

### 4.2.3. The Transformation to Practical Utilitarianism during the Decline Phase of Buddhist Architecture Development

From the Yuan Dynasty (1271–1368) onward, the growth of historical Buddhist architectural samples began to decline, marking a shift from its earlier flourishing state to a period of decay. The most prominent factors affecting its distribution and architectural form include institutional changes under the influence of minority regimes, the evolution of Buddhist artistic styles under the influence of ethnic cultural integration, the incorporation of Daoist deities and local folk gods into the Buddhist belief system, and local economic growth driven by the incense-market activities.

The Yuan Dynasty stands out as the inaugural Chinese feudal dynasty founded by a nomadic tribe, with rulers hailing from the Mongolian steppes and adhering to Tibetan Buddhism. In the Zhejiang region, the development of Chinese Buddhism, predominantly in the form of Chan, continued the Five Mountains and Ten Monasteries system from the Song Dynasty. The newly added Buddhist architecture samples primarily concentrated around Lin'an 临安 in Hangzhou. On one hand, the ruling class established an administrative center for Chinese Buddhism in Hangzhou for the Jiangnan region, and granted Tibetan Buddhist monks the authority to administer the Buddhist temples, monks, and organizations within Zhejiang. This was a revolutionary move against the Buddhist administrative system of previous dynasties, and led to a precipitous drop in the number of new historical Buddhist architectural samples, and many monks, in a state of passive resistance, either secluded themselves in mountains or fled to Japan, with numerous temples destroyed during the warfare at the end of the Yuan Dynasty.

On the other hand, within the context of the multicultural amalgamation during the Yuan Dynasty, Buddhist architectural art, through the assimilation and re-innovation of Tibetan Buddhist art, forged a Han-Tibetan fusion style (Liu 2023, pp. 156–62), representing the most prominent characteristic in the evolution of Buddhist architectural art styles of this period. The most representative of this are the Yuan Dynasty rock carvings at Feilai Peak in Hangzhou, due to the main benefactor being Yang Lianzhenjia 杨琏真伽, who was from the Western Xia 西夏, and also the commander of Han Buddhism in the Jiangnan area. The combination of the traditional Chinese Buddhist iconography with the official endorsed Tibetan Buddhist sculptural style, termed "Western Paradise Brahman Features"[22] (Song 1976, p. 4547), stemmed from two driving factors. One was the alignment of Tibetan Buddhist art and Western Xia art with the mainstream societal religious practices and aesthetic preferences, and the other was the control over the management rights of Buddhist architecture. Not only was there a blend between the traditions of Han Buddhism and the

official Tibetan Buddhism, but also incorporated were the Western Xia Buddhist art style inherited from the Dunhuang 敦煌 Buddhist art style.

The Buddhist sculptural art of Feilai Feng harmoniously integrates with the Western Heavenly Brahmanic style, evolving the facial features of the statues to exhibit full, rounded cheeks. Male Buddhist statues tend to be more plump or rounded, while the female Buddhist statues have subdued feminine characteristics, tending to lean towards an elegant and dignified appearance. For example, the statues within the 84th niche of Feilai Peak, exemplifies the Western Paradise Brahman Features style, with their light and close–fitting attire and connected eyebrows. The Buddha Mother's slender and slightly closed eyes, rounded chin, and the nearly upright posture of the bodhisattvas on both sides, exemplify the style of Chinese Buddhist statue art.

Additionally, the clothing style of the statues is either represented with bare shoulders and tight, thin attire, or bare upper bodies, with only a tight skirt worn below, demonstrating Tibetan style. This stands in stark contrast to the flowing and weighty robes worn by Southern Dynasty scholar officials, where complicated folds and patterns coexist. For instance, the four-armed Avalokiteshvara in the 93rd niche is only exposed at the chest, with the kasaya having drooping double collars and intricate folds, highlighting the pursuit of the beauty of clothing lines found in Chinese literati paintings. The characteristic "wrathful appearance"[23] of Tibetan Buddhist statues has also been diminished in the integration of Han and Tibetan styles. For example, the Vajrapani bodhisattva in the 32nd niche retains the fierce expression with wide open eyes, but the rounded chin, abdomen, and short, chubby stature render the statue more innocent and gentle. The attire of the statues also presents in the artistic style of Tibetan Buddhism, characterized by baring the shoulders and clinging lightly to the body, or being bare-chested with only a fitted skirt worn below. Additionally, the attire of the statues also reflects the robe style of the Southern Dynasty's scholars, which is loose, heavily draped, with complex folds and patterns.

Starting with the Ming Dynasty, the renowned "Four Famous Mountains"[24] supplanted the Five Mountains and Ten Temples system, with Mount Putuo 普陀, located in the eastern coastal area of Zhejiang, and emerged as the new center of Buddhist culture, marking a complete shift of Buddhist centers from cities to the mountains and rural areas. There are two most prominent features of this period. Firstly, influenced profoundly by Daoist thought and folk beliefs, both Buddhist architecture and worship activities gravitate towards simplicity and unpretentiousness. A form of "Syncretic Buddhism" (Zhang 2002, p. 20) emerges among the people, characterized by a blended worship of deities from Buddhism, Confucianism, Daoism, and folk religion. The overall environment of the mountain temples leans more towards Daoist inclinations, emphasizing the aesthetic of mountain and forest landscapes, while rural temples integrate the daily lives and folk beliefs of the rural residents, allowing Buddhas and folk deities to coexist. The phenomenon of having a Buddhist temple in each village is prevalent in Zhejiang. Taking Wucheng 乌程 in Zhejiang as an example, there are 62 Buddhist temples in the county town, and over 40 Buddhist temples in villages outside the town (Yang 2021, pp. 255–267). Buddhist statues are also worshipped in the homes of rural residents.

Capitalizing on the convergence trend of the three teachings and by integrating with folk beliefs and people's practical needs, Buddhism has formed a secularized representation of Pure Land faith. This allowed Buddhism to deeply infiltrate the daily production and life of rural residents, blending with local clan systems, gentry systems, land systems, and especially with folk beliefs in ghosts and deities. This not only enhanced the utilitarian aspect of Buddhist activities but also constituted a conglomerate of various beliefs in the rural areas of Zhejiang. Consequently, the amalgamation of folk Buddhist temples with other folk temples has further deepened trends toward popularization, secularization, and pragmatization. The architectural layouts also display a mixture of different sectarian and religious architectural styles, such as the incorporation of the Vinaya Hall from the Vinaya School in the hall arrangement of Chan Buddhist temples. Particularly significant is Buddhism's proactive alignment with Neo-Confucianism, which has further unified the

three religions and influenced the worship of deities in Buddhist temples. Influential folk deities join the ranks of temple guardians, reflecting a characteristic of coexistence among different religions. For example, the Longquan 龙泉 Temple in Yuyao 余姚, Zhejiang, has the Dragon King from Daoism, and the temples in Mount Tiantai 天台, revere Guan Yu 关羽 as the temple guardian (Xu and Sima 2014, p. 95). Moreover, to express gratitude to benefactors, temples would incorporate benefactors into their worship system, such as the Shang Tianzhu 上天竺 Temple in Hangzhou, to establishing a shrine in gratitude to the benefactor Liang Yao 梁瑶 (Cao 2011, pp. 53–64).

Secondly, the expansion of incense markets drove the mutual growth of temples and local economies, transforming Buddhist temples into venues for folk activities and commodity trade, marking a shift from the sacred to the secular. Emerging during the Ming and Qing Dynasties, incense markets represent commercial trading and celebratory activity organized by the common populace around Buddhist temples during festive seasons (Zhang 2018b, p. 9), which exemplified a period where Buddhism became further secularized and Buddhist rituals evolved into folk customs. Incense markets are typically of two types (Cai 2010, pp. 24–29). One type is a commercial place for trading various materials, for making incense, such as finished incense and candles, as well as handicrafts, daily necessities, and food produced by the local people, such as the West Lake 西湖 incense market in Hangzhou 杭州 (Zhang 2018a, p. 104), which operates annually from the Hua Zhao 花朝 Festival[25] to the Dragon Boat Festival. The other type is a temple fair-styled temporary market, such as the Anguo 安国 Temple incense market in Haining 海宁 (Wu 2015, p. 155), which only takes place on June 29th of the lunar calendar, which is the day commemorating the birthday of Guanyin 观音.

The popularity of the incense markets in Zhejiang is primarily attributed to the abundance of Buddhist temples, a large community of Buddhist devotees, a flourishing commercial economy, and convenient waterway transportation. Since the Three Kingdoms period, Hangzhou has been dubbed the Buddhist Kingdom of the Southeast. By the time of the Ming and Qing dynasties, the number of Buddhist temples had soared to over 2000 (Xiao 1996, p. 94). A stable social environment facilitated population growth; in Hangzhou alone, the population increased from 280,000 people in the early Qing Dynasty to 2.7 million people in the mid–Qing Dynasty (Li 1974, pp. 1195–1196). The prosperous sericulture and handicraft industries during the Ming and Qing dynasties propelled the prosperity of commerce and industry in Zhejiang. Sericulture constituted a primary income source for the local people, and the timing of the entire production activity coincided with the West Lake Incense Market. Consequently, sericulturists combined the folk customs of praying for successful silkworm rearing and abundant silk harvest with the incense-market activities, further expanding the scale of the Buddhist devotee community. The well-developed water-transport system made travel convenient, drawing visitors from all over the country. Large-scale incense markets, such as the West Lake Incense Market, attracted tens of thousands of people (Cai 2010, pp. 24–29). Such dynamics not only intensified the population mobility between rural and urban areas but also promoted the sale of rural handicrafts.

Incense–market activities provided economic support for Buddhist temples. Monks generated funds for the maintenance and construction of temples and for supporting religious activities through the sale of Buddhism-related goods and the substantial monetary offerings from devotees during worship and incense reverence (Fan 1989, p. 9). Concurrently, the activities of the incense markets integrated Buddhist temples with the market economy, attracting participation from groups of people with diverse backgrounds and beliefs, promoting the social integration of the temples. During the incense-market period, women participated in Buddhist activities, temple visits, and trading activities, which also demonstrated the open views of a feudal society under the influence of Buddhist culture in the specific social context.

## 5. Conclusions and Discussion

### 5.1. Conclusions

This study focuses on the rich historical and cultural resources of Buddhist architecture in Zhejiang. Using cultural geography as a foundation, this study considers 159 representative historical Buddhist architectural sites. By analyzing the geographical distribution and historical evolution of these samples, it explores the clustering characteristics and quantitative changes in Buddhist architecture in Zhejiang during different historical periods. Furthermore, this study explores the distinctive spatial-distribution characteristics of different developmental phases of Buddhist architecture in Zhejiang. Various factors, including naturel elements, transportation, religious and cultural transmission, politics, construction techniques, and economic dynamics, shape these characteristics. Notably, the rise and decline of Buddhism and the trend towards secularization are the most prominent driving forces affecting the spatial distribution of Buddhist architectural, fostering diversity, practicality, and integration in form, function, and aesthetics of Buddhist architecture.

1. The spatial distribution of representative historical Buddhist architecture in Zhejiang reveals pronounced geographical differentiation patterns, with numerous historical architectural sites distributed in the northern and southeastern coastal regions and fewer in the western part. The overall distribution is characterized by a single-core clustering, coupled with temple cluster networks radiating from the core to the periphery. The concentrated area in the center is located in the central urban area of the northeastern part of Hangzhou, historically the economic, cultural, and artistic hub of Zhejiang. The distribution of different types of historical Buddhist architectural sites is relatively uneven. Apart from the Buddhist temples, which exhibit a single main core with a multiple secondary cores clustering pattern, the sample types of other categories all show a single-core clustered distribution.

2. The spatial-movement trend of mean-center migration of representative historical Buddhist architectural sites aligns with the shifts in political and economic centers throughout various historical periods in Zhejiang. This alignment is primarily influenced by the phase-wise changes in the dissemination of Buddhist culture, the institutionalization of Buddhist by the ruling class, and the formation and development of different Buddhist sects. The movement of the mean center of historical Buddhist architectural samples exhibited four distinct directional phases, overall shifting from north to south. The spatial distribution of representative historical Buddhist architectural sites in Zhejiang during different historical periods, shows a trend shifting from random to clustered, then back to random, and eventually to dispersed, with the growth trend in sample quantity showing clear phases. Particularly, the period from the Southern Dynasties to Song Dynasty is the most crucial phase in the development of historical Buddhist architectural sites, establishing a network of temples radiating from Hangzhou to the entire province.

3. Natural and transportation factors have a significant interplay with Buddhism in Zhejiang, shaping the spatiotemporal distribution and evolution of Buddhist architecture sites. The areas with concentrated sample distribution tend to showcase a superior natural environment, convenient transportation, and high intensity of human activities. These areas are also a comprehensive reflection of a higher level of economic and social developments, and are of profound Buddhist cultural heritage. The variety in terrain of Zhejiang has given birth to distinct urban Buddhist temples, rural Buddhist temples, and mountainous Buddhist temples. Both urban and rural temples are mostly located in plain areas with extensive waterways, displaying a marked affinity for water, with 90% of samples clustering in cities and surrounding rural areas that have convenient transportation and developed economies. In contrast, mountain temples constitute only 10% of the total samples, closely integrating with the natural environment and exhibiting characteristics of landscaping. With the development of transportation and urban expansion, the accessibility of water and land transporta-

tion links historical Buddhist architectural sites distributed across the province into a temple network centered around Hangzhou. This directional preference in site selection of Buddhist architecture in Zhejiang is particularly evident during the Ming and Qing dynasties. Temples located along the Jiangnan Canal and at the transportation hubs on the eastern coastline often attracted numerous pilgrims during the incense-market events, highlighting the enhancement of the temples' influence due to their accessibility and convenience in transportation.

4. Feudalism, centralized power, and Confucian thought shaped the hierarchy of historical Chinese society. The agricultural economy and market economy interwove to form a composite economic structure. All these factors had a significant impact on the distribution and development of Buddhist architecture across different historical stages. With changes of dynasties and capital migrations, coupled with the southward movement of the population, the number of monks in Zhejiang increased. This bolstered the rapid dissemination of Buddhist culture and architecture. This, in turn, led to growth in the number of devotees, thereby stimulating the demand for Buddhist activity venues and the vitality of the temple economy, and subsequently promoting the privatization of temples. Under centralized rule, the politicization of Confucian thought intermingled with Buddhism, facilitating the Sinicization and hierarchization of Buddhist architecture. This has resulted in two distinctly different Buddhist architectural systems, official and folk, increasing the dependency of Buddhist structures on the ruling class. However, the phenomenon of hierarchical Buddhist architecture contrasts with the Buddhist philosophy of universal equality and harmonious integration, leading to a longing among monks and the literati class to distance themselves from the secular world, becoming one of the significant considerations in choosing locations for mountain temples. In an economy dominated by agriculture, land donations from the ruling class and mainstream social groups aided Buddhist temples in acquiring and expanding their land holdings. The highly developed temple economy provided a stable economic foundation for the existence and development of Buddhism. This facilitated an increase in the amount of Buddhist architecture, and further solidified the social status of Buddhist architecture. Additionally, the activities of the incense market developing around the temples stimulated the flourishing of local handicraft and the commercial economies, which increased population mobility, and closely integrated temple activities with folk customs, and strengthened the temples' function as tourist attractions.

5. The integration and expansion of Buddhism in Zhejiang, especially its interactions with Daoism, Confucianism, and folk beliefs, have had a decisive impact on the spatiotemporal distribution of Buddhist architecture. Before Buddhism's introduction, Zhejiang already had a rich tapestry of religious beliefs and cultural traditions. This backdrop facilitated the fusion and adaptation of Buddhist culture with the local culture and belief systems, providing fertile ground for the secularization and diversification of Buddhist architecture. The trend of syncretism among the three teachings and their spatial proximity further facilitated interactions among them in cultural and religious activities. This synergy led to three distinctive characteristics in Buddhist architecture: Firstly, the multiplicity and diversity in the forms of Buddhist architecture. Given the political and social demands of the ruling class and mainstream groups, combined with the Buddhist educational and economic needs, official Buddhist architecture sites became part of the traditional Chinese architectural paradigm, evolving from standalone structures to large-scale architectural complexes that combine central courtyards with multiple subsidiary courtyards. In terms of layout, form, and structure, they closely resemble palaces and traditional residences in the Zhejiang. Secondly, Buddhist architectural functions have evolved towards practicality, integrating a range of uses. To better adapt to local culture and lifestyles and better serve the community, Buddhist temples expanded their roles, encompassing educational exchange, commercial trade, festive entertainment, and social gatherings, thus

enhancing the utilitarian function of Buddhist activities. Moreover, in the context of the convergence of the three teachings, Buddhist temples incorporate Daoist and Confucian deities and local gods of folk belief as objects of worship, reflecting the convergence and integration in worship functions under the trend of syncretism.

6.  Lastly, there has been a notable popularization and integration of the aesthetic forms of Buddhist architecture. Interaction between Buddhism, folk culture, and local beliefs have rendered the styles and decorative features of Buddhist architecture more accessible and understandable, resonating deeply with the daily lives and spiritual needs of the public. As Buddhist architecture integrates with local religions in doctrines and beliefs, it assimilates and merges artistic styles and elements from varied cultures, displaying characteristics of openness and inclusivity.

In general, natural elements, transportation, feudal systems, economic structures, and indigenous religious thoughts and cultural traditions have collectively influenced the spatiotemporal distribution and evolution of Buddhist historical architecture sites. The interplay between Buddhist architecture and these diverse factors reflects a profound process of deep integration and reciprocal adaptation. The unique geographical location and abundant natural resources in Zhejiang have provided Buddhist architecture with exceptional conditions for development. Historically, as a center of politics, economics, and culture, Zhejiang has provided substantial economic support for the construction and preservation of Buddhist architectural sites. These sites represent not only the manifestations of Buddhist culture but also a fusion of Confucianism, Daoism, and local folk beliefs. With the advancement of technology and cultural exchange, Buddhist architecture has transitioned from its initial, singular Indian Buddhist architectural style to encompass a diverse range of forms rooted in traditional Chinese architectural styles. Simultaneously, their functions have expanded beyond serving merely as venues for religious activities, taking on multiple roles in education, social interaction, economy, and entertainment.

### 5.2. Discussion

The spatiotemporal distribution and evolution of Buddhist architecture in Zhejiang is a multifaceted and multilayered complex process intimately linked with the region's history, culture, economy, and political tapestry. It reflects the transitions and developments of Buddhism across history, culture, economy, and society. Based on this, studies and practices on historical Buddhist architecture in Zhejiang should also focus on the following aspects:

1.  Efforts should be made to actively conduct theoretical research on the historical Buddhist architecture in Zhejiang, to uncover its cultural genes, and to construct a discourse system that aligns with the local historical and cultural characteristics of Zhejiang. This research will provide theoretical support for the effective protection of historical Buddhist architecture in Zhejiang, while enhancing its international influence.

2.  Consideration should be given to the holistic preservation of the cultural ecosystem of historical Buddhist architecture in Zhejiang. The Buddhist architectural sites in Zhejiang, from different historical periods, have been influenced by the succession of dynasties, the development of Buddhism, and environmental factors, and has manifested in specific clustered regions and distribution patterns. For effective preservation, it is crucial to recognize the integrality of its cultural ecosystem of historical Buddhist architecture. With the consideration of various influencing factors, such as the terrain, rivers, transportation, political system, the spread of Buddhist culture, and regional culture, there should be a formulation of differentiated, targeted, and sustainable protection strategies and plans. Furthermore, the protection and utilization of historical Buddhist architectural sites need to refer to the relevant regulations for the preservation of immovable cultural relics, acknowledge its economic function and social development of the region, and consider the practical issues such as the

functions, spatial transformation, and management of Buddhist historical buildings in the contemporary tourism and cultural context.

3.　However, this study still has some limitations. For example, in the selection of historical Buddhist architectural sites in Zhejiang, different identification criteria can often bring certain deviations in the research results, necessitating more detailed investigation for further research. For a more profound insight, future research should focus on a more specific and micro-level study on the spatial distribution of Buddhist architecture during different historical periods and influencing factors. Delving into ancient texts that document the layout of these structures will be a pivotal avenue to explore. In particular, exploring the social-connection features between historical Buddhist architectural sites and historical Buddhist figures, constructing networked spatial structures, and proposing spatial-integration-protection strategies for historical Buddhist architectural sites are crucial.

**Funding:** This research received no external funding.

**Institutional Review Board Statement:** Not applicable.

**Informed Consent Statement:** Not applicable.

**Data Availability Statement:** Not applicable.

**Conflicts of Interest:** The author declares no conflict of interest.

## Notes

[1]　Kuaiji 会稽, the administrative center of Jiangnan, and the earliest organizational system in Zhejiang, belonged to the "Kuaiji Province, Zhang Province and Minzhong Province" among the 36 provinces set up by Qin Dynasty (B.C. 220–207), with a total of 15 counties. In the Western Han Dynasty, Zhejiang belonged to Kuaiji Commandery and Danyang County, with 20 counties in total. In the Eastern Han Dynasty (B.C. 202–A.D. 8), it belonged to Kuaiji Province, Wu Province and Danyang Province, with 23 counties in total. During the Six Dynasties (222–589), under the influence of frequent dynastic changes, the organizational structure of Zhejiang continued to change. During the Three Kingdoms period, a total of 6 provinces and 44 counties were established, while during the Western Jin period, a total of 6 provinces and 49 counties were established. During the Eastern Jin period, a total of 7 provinces and 51 counties were established. During the Southern Dynasty, it belonged to East Yangzhou, with 5 provinces and 23 counties under its jurisdiction, under the jurisdiction of Shaoxing and Ningbo.

[2]　Kang Senghui 康僧会 is from Sogdiana, in present-day Uzbekistan. In the tenth year of Chiwu's reign (248), Kang Senghui went north from Jiaozhi State to Jiangnan, built huts, set up statues of Buddhas, and translated scriptures for preaching (Shi 1992, p. 15), which had the most profound influence on the spread of Buddhism in Zhejiang Province.

[3]　Feilai 飞来 peak, located in a mountain between Lingyin 灵隐 and Tianzhu 天竺 in Hangzhou 杭州, Zhejiang. Huili 慧理, an Indian monk saw this peak, "This is the small ridge of Tianzhu Spirit Mountain, I don't know why I flew here?" Therefore, it was named Flying Peak (Guan 2020, p. 32).

[4]　https://www.chinabuddhism.com.cn/zdsy/95/ (accessed on 18 August 2023).

[5]　http://www.zjfjxh.com/Public/NewsInfo.aspx?type=1&id=7bad33c7-0297-4f76-b1e0-874dfdbf3eeb (accessed on 18 August 2023).

[6]　http://wwj.zj.gov.cn/art/2020/6/12/art_1639081_43375775.html (accessed on 18 August 2023).

[7]　http://wwj.zj.gov.cn/art/2020/7/23/art_1639081_52227993.html (accessed on 18 August 2023).

[8]　There were mainly nine ancient post roads: Suzhou 苏州 to Hangzhou 杭州, Hangzhou to Fuzhou 福州, Hangzhou to Mingzhou 明州 (Ningbo 宁波), Muzhou 睦州 (Jiande 建德) to Wenzhou, Hangzhou to Xuanzhou 宣州 (Jiangsu 江苏 and Anhui 安徽), Hangzhou to Huizhou 徽州 (Anhui 安徽), Yuezhou 越州 (Shaoxing 绍兴) to Wuzhou 婺州 (Jinhua 金华), Yuezhou to Taizhou 台州, and Mingzhou to Wenzhou 温州.

[9]　By the end of 2021, first-class and second-class highways in Zhejiang, the total length of the first-class highway was 8105 km, and the total length of the second-class highway was 10,860 km.

[10]　Shanyin 山阴 is a historical name for an old county, which is now part of Yuecheng District and Keqiao District in Shaoxing 绍兴, Zhejiang.

[11]　Yongxing 永兴 is a historical name in Xiaoshan 绍兴, Hangzhou 杭州, Zhejiang, and was part of Kuaiji 会稽 Commandery.

[12]　Hui Li 慧理, a Indian monk in the West of Tianzhu 天竺. Since Huili successively established five temples in Lingyin 灵隐 and Tianzhu 天竺 Mountain, such as Lingyin 灵隐 Temple and Lingjiu 灵鹫 Temple, he was honored as the "Lingzhu Founder" by later generations.

[13] The monastic official system in Zhejiang originated from the system of the Later Qin Dynasty (384–417), through the establishment of "Sengzheng 僧正", "Yuezhong 悦众" and" Seng Lu 僧録" (Shi 1992, p. 240) to gradually manage the central and local monks and nuns in states and counties and handle daily affairs. The monks were in charge of grassroot temples such as the "Fazhu 法主" (Shi 1992, p. 313) to assist the ruling classes in managing and liaising with the vast number of monks and nuns.

[14] The Buddhist sects in the Tang Dynasty include Sanlun 三论宗, Tiantai 天台宗, Huayan 华严宗, Faxiang 法相宗, Chan 禅宗, Esoteric 密宗, Ritsu 律宗 and Pure Land school 净土宗.

[15] In the Tang Dynasty government code of Tang lv Shu Yi 唐律疏议, it is clearly stipulated that the "three principles 三纲" are the upper seat, temple master and Duvina (Liu 1996, p. 528). The "Three principles" of high-level Buddhist temples are ordered by the emperor, while local Buddhist temples and general Buddhist temples are appointed by Jiedushi 节度使 and state officials, and need to be reported and filed (Zhang 1997, p. 366).

[16] In Sanskrit, "aranya" refers to a secluded place suitable for the meditation and residence of ancient Indian monks, or a Buddhist temple where monks gather.

[17] The Five Mountains of the Chan sect temple include Jingshan Temple in Yuhang, Lingyin Temple in Qiantang, Jingci Temple, Tiantong Temple and Yuwang Temple in Ningbo. The Ten Temples of the Chan sect temple include the Zhongzhu Temple in Qiantang, the Daochang Temple in Huzhou, the Jiangxin Temple in Wenzhou, the Shuanglin Temple in Jinhua, the Xuebao Temple in Ningbo, the Guoqing Temple in Taizhou, the Xuefeng Temple in Fuzhou, the Linggu Temple in Jiankang (Nanjing), the Wanshou Temple and the Huqiu Temple in Suzhou.

[18] *Ying Zao Fa Shi* 营造法式 is an architectural treatise written by Li Jie 李诫 during the Song Dynasty. It was an official publication that provided design and construction standards for architecture. This book clearly defined various design standards, regulations, materials, construction quotas, and indicators, establishing a hierarchical system for housing construction, artistic forms of architecture, and strict guidelines for materials and construction techniques.

[19] Daoism advocates for a retreat from the noisy commotion of urban life, emphasizing harmony with nature and closeness to tranquil and serene natural surroundings.

[20] Confucian culture spirits of transcending and engaging with the world. On one hand, Confucianism emphasizes active participation in society and fulfilling social responsibilities. On the other hand, it also focuses on personal cultivation, self-reflection, and spiritual development.

[21] The seven halls consist of the mountain gate, Buddha hall, Dharma hall, monk's room, canteen, bathroom, and toilet, with the Dharma hall, Buddha hall, and mountain gate located along the central axis, and the other buildings as annexes.

[22] The style of Western Paradise Brahman Features originate from East India and possess physical features characteristic of the Indian people, such as large, protruding eyes, prominent upper eyelids, thick lips. Male Buddhist statues have a robust and muscular physique with a noticeably narrower waist, while female Buddhist statues exhibit distinct feminine traits, with rounded and graceful curves.

[23] The wrathful appearance is one of the types of deities' facial expressions in Tibetan Buddhism, typically characterized by wide-open eyes and raised wrists, presenting a fierce and angry image that can instill fear in the viewer.

[24] The Four Famous Mountains are the ashram for the four Bodhisattvas of Chinese Buddhism to show their sainthood and promote Buddhism. Mount Wutai 五台 in Shanxi 山西 is the ashram for Manjushri Bodhisattva, Mount Putuo 普陀 in Zhejiang is the ashram for Guanyin 观音 Bodhisattva, Mount Emei 峨眉 in Sichuan 四川 is the ashram for Puxian 普贤 Bodhisattva and Mount Jiuhua 九华 in Anhui 安徽 is the ashram for Kṣitigarbha.

[25] Hua Zhao 花朝 Festival is celebrated in the lunar month of February every year. It is considered the birthday celebration of flowers in China and has been popular since the Song Dynasty, particularly in the Jiangnan region. During this festival, girls gather to worship the Flower deity, burn incense, and engage in various activities such as going on outings, planting flowers, making flower cakes, and enjoying each other's company.

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
