# Peer review of "Gathering Southward under Secularization and Syncretism: Study of the Spatial-Temporal Distribution and Influencing Factors of Chinese Historical Buddhist Architecture in Zhejiang"

_religions, doi:10.3390/rel14111398_

Round 1
Reviewer 1 Report (Previous Reviewer 3)
Comments and Suggestions for Authors
I have observed that the author has made considerable revisions to the paper. While some of these changes are commendable as they make the writing more relevant to the specific architectural aspects of Buddhism in Zhejiang province rather than discussing Buddhism in a broader context, there are still two notable weaknesses that persist:
1. The text continues to place more emphasis on temporal changes rather than spatial ones in historical Buddhist architecture.
2. Among the four conclusions presented in the paper, it appears that the author has primarily discussed, at most, two aspects. In fact, the summary of "Overall, it exhibits a "single core clustered", coupled with temple cluster networks radiating from the core to the periphery. The concentrated area in the center is located in the central urban area of northeastern part of Hangzhou, historically the economic, cultural, and artistic hub of Zhejiang. The distribution of different types of historical Buddhist architectural sites is relatively uneven. Apart from the Buddhist temples, which exhibit a "single main core and multiple secondary cores clustered", the sample types of other types all show a "single core clustered" characteristic" should really be elaborated in the main text. Unfortunately, those key words are rarely seen in the writing. The last conclusion "where agricultural economy, bureaucratic economy, and market economy interwove to form a composite economic structure" is not much seen in the discussion either. What can be seen is only limited to the incense economy.
3. Finally, I do not understand why the author has four discussions in the end. I do not see their close connection to the text.
Comments on the Quality of English LanguageNeeds to be improved. For instance, in the abstract, it has such expressions, "in specifically..." Also, there are quite a few places, the punctuations are misused.
On page 23, the whole paragraph "under the dual drive of power and utilitarianism..." appears twice.
Author Response
Please see the attachment.

Reviewer 2 Report (New Reviewer)
Comments and Suggestions for Authors
This is a very well researched paper, that contributed to our understanding of Buddhism in Zhejiang province. Thank you very much. The paper contains a number of minor problems. Please see my suggestions concerning them below:
religions-2677487-peer-review_comments
Lines 5-6
Change “the architecture of Chinese Buddhist” to “Chinese Buddhist architecture”
Line 8
The text “in cluster” is unclear. Please omit or revise
Change “culture” to “cultural”
Line 13
Change “clustered” to “clustering”?
Line 51
Change “Buddhism” to “Buddhist”
Line 55
Change “political,” to “political”
Line 73
Correct “architecturl” to “architectural”
Change “the monk” to “monastic”
Line 85
Change “dwelling of monk” to “housing of monastic”
Line 94
Correct “Sangharama” to “Saṅghārāma”
Line 97
Correct “Stupa” to “stūpa”
Line 115
Correct “Dynastes” to “Dynasties”
Line 121
Change “periodsaw” to “period saw”
Line 144
I’m not sure it’s correct to say that “Tibetan Buddhism prevailed”. Perhaps you might qualify this, saying that it prevailed among the ruling class?
Line 148
Change “distinct” to “a distinct”
Line 150
Change “declined” to “decline”
Line 153
Delete text “a trend of”
Line 159
The word “sensitive” does seem appropriate in this context. Perhaps replace it with “prevalent”
Line 189
Change “post” to “following”
Line 195
Change “can reflect” to “reflects”
Line 356
Change “cener” to “center”
Line 435
Change “Dynastyto” to “Dynasty to”
Line 611
Change “initially” to “was initially”
Line 624
Remove commas in “large-scale, landscaping,”?
Line 629-630
The text “ceremonial hierarchical” is unclear. Please rephrase.
Line 634-635
The text “varied building systems of different hierarchical system” is unclear and redundant. Please rephrase.
Line 793
Change “for the” to “the”
Line 818
Correct “sanghrma” to “saṅghārāma”
Line 1095
Rephrase “needs of Buddhist education and economics” to “Buddhist educational and economic needs”
Line 1128
Please explain what “Indian Garan style” means.
Comments on the Quality of English Language
The English is basically fine but exhibits some minor problems, as I noted above. Please edit for grammatical issues.
Author Response
Please see the attachment.

This manuscript is a resubmission of an earlier submission. The following is a list of the peer review reports and author responses from that submission.
Round 1
Reviewer 1 Report
Comments and Suggestions for Authors
This is a wonderfully detailed and highly informative study of the evolution of Buddhist architecture in the southern region of China, Zhejiang. This well- rounded examination considers in depth the various factors that determined the development of different types of architecture in the area -- geographical, political, historical cultural and religious factors. This essay adds much to our knowledge of different types of Buddhist architecture, their function and evolution in Zhejiang.
Author Response
Response to Reviewer 1 Comments:
I am very grateful to Reviewer for your recognition of my article. In future studies, I will investigate ancient Buddhist architecture in more detail, dig into the records of the distribution of Buddhist architecture in relevant ancient books, and conduct more specific and micro studies on the spatial distribution and influencing factors of Buddhist architecture in historical periods, especially the relationship between ancient Buddhist architecture and economic, population, technology and other factors.
Thank you again for your positive comments on my manuscript. Your comment is valuable and helpful for improving my research direction.
Best regards!
Yours sincerely,
Reviewer 2 Report
Comments and Suggestions for Authors
As noted in my attachment, this paper requires a thorough revision. It should not have been submitted in this form. Main points:
thesis statement unclear
mixes up contemporary and historical areas of discussion, which should be 2 separate papers
technical aspects of geolocation analysis are too long and should be summarized
lacks any explanation of why the results are this way
I suggest the author take the conclusion section and use that to organize the paper. Add technical charts/diagrams only when they help illuminate points. Focus on historical issues only--do contemporary analysis in a separate paper. And reformulate the thesis statement. Overall the paper can be reduced by 40-50% and still be sufficient. Some interesting points are touched on but most of the paper is dull.
And fix the many English issues.

Verb tense needs to be reviewed. Plurals are not always used correctly. Words are capitalized when not necessary. Also quotation marks are used too much. Some sentences are confusing and should be eliminated or in some cases rewritten. But eliminating is best. Needs a good English editor before submission.
Reviewer 3 Report
Comments and Suggestions for Authors
This paper uses the Spatial analysis method of ArcGIS to study the spatial-temporal distribution and influencing factors of Chinese Buddhist architecture in Zhejiang Province. I think the research makes contribution to the field. However, it also needs considerable improvements:
1) The literature review needs to be changed by telling what the researchers have done and how they have brought changes to the field. For instance, on page 4 between lines 193and 195, the author states that “Based on the spatialtemporal analysis function of GIS and set the temple as the example, Xu Ying makes research on the geographic distribution of the temples of the sui and tang dynasty.” What’s the relevance/significance of Xu Ying’s research? Also, the literature review needs to relate itself to the current research.
2) When the author discusses the spatial distribution of Buddhist architecture, s/he cites a lot of names of the locations. To readers who are not familiar with the area, what are the meanings of these places? Rather, the author should analyze the geographical characteristics of the concentrated area. Otherwise, the proper names do not make sense.
3) Regarding the discussion of the temporal variation, the line on 418-419 “and the overall movement distance shows a trend of decreasing first and 419 then gradually increasing” sounds unclear to me.
4) Although the author concludes that social, political, cultural, and technological factors influence the spatial-temporal distribution of Chinese Buddhist architecture in Zhejiang Province. I rarely see the discussion of economical and technological factors. In fact, the space the author allows for the analyses of this area is also limited compare to the large proportion of other sections.
5) The English writing needs to be improved. Some sentences have grammatical mistakes. For instance, the sentence between 217-219 is not correct. The author also needs to pay attention to the capitalization and small cases of words.
Comments on the Quality of English LanguageThe English language needs to improved.
Round 2
Reviewer 2 Report
Comments and Suggestions for Authors
56 “The term Buddhist temple”: which Chinese term? Please note it.
58 Which Chinese term for “buddhist temple?” Pleas note it.
68 change promoting to promoted
80/81 Need transliteration of all Chinese terms, please.
130 eliminate “the” before pure land thought
133 Considering the teachings. This sentence is too long and confusing. Revise into several shorter sentences.
139 eliminate “of the trend”
148 During the Ming dynasty…
149 Incense Market: no capitals, please
149 and organizing of the….
148 Make two sentences: around Buddhist architectural structures. These activities including during incense…
170 The Geographical Information System
184 eliminate some aspects of
187 change believed that to found that
234 in “national…”: suggest you say “in the following studies: National Buddhist temples…”
237 They: What is “they?” Please be specific. Do you mean the architectural sites discussed in the reference works?
242 you cannot say the architecture. You must say the architectural site, or architectural structure, or architectural sample, etc.
244 same as above
248 say the existing Buddhist and represnetative ancient architecture examples, or say map of existing…. You cannot say “the architecture” except in an abstract sense.
249 same problem
297 same problem
299 Yes! You did this correctly! Examples
306 Also correct
314 ancient Buddhist architecture: This is ok because you didn’t say “the.”
340 Republic of China period cannot be considered ancient, since it is modern. Does your study include from Three Kingdoms to ROC? If so don’t call it modern. Call it historical to distinguish it from contemporary. In fact you should make a note of the time period you study when you introduce your data sources. Just say “This study look as historical sites built from the Three Kingdoms period through the Republic of China period.” Something like that.
354 So all your use of “ancient” needs to be revised. (I note you did not mention “ancient” in the title, and you don’t start to use that word until around line 230. Please rivise it to be “historical” or simply eliminate the adjective.
428 growth in the number of ancient Buddhist architectural sites….
436 eliminate a in a growth
481 The water source, as one of the main factors affecting the spacial distribution: eliminate the first and third “the”
504 eliminate last the
510 eliminate 7
536 again, I strongly urge you to use Daoism instead of Taoism, which is old-fashioned
551 presented itself in two forms
561 the Chinese culture: eliminate “the”
564 It: What is “it?” Be specific.
568 created
578 The Buddhist architecture transformed from the house: Not clear. Say something like”. Buddhist architecture influenced by residential styles” or something similar.
583 Increase in: say “rise of”
585 the practice-oriented: eliminate the
596 Emergence of vihara: This is confusing. Is the vihara a new kind of Buddhist architecture in this period? A new way of organizing the temple? What period? You have not used “vihara” before. This is sudden and requires explanation. Revise this paragraph.
598 the mountains of
601 and built a vihara; again, why are you using the Sanskrit word? Why not use the Chinese term? You have consistently used Chinese terms.
605 for example: This should be a new sentence: For example,….
615 the Buddhist temples: eliminate the
617 the mainstream social groups: eliminate the
618 Idolatry core: what is that? Why is idolatry capitalized? When you introduce a concept, define it.
622 the Emperoro of Buddha statues: What does this mean? Unclear.
624 the Buddha statues: eliminate the
628 the large-scale; delete “the”
633 Kuaiji became the center of a dense…
641 The changes…constitute a significant turning point…
645 were presented as: change to “took the form of”
646 the trend of sinicization: please define this term somewhere.
651 The architectural forms used in these settings consisted of numerous ups and downs…
653-4 The emergence of….was influenced by
656 the paradise: delete the
656 On the other hand: make this a new sentence.
657 It is also closely related to something inherent in Chinese culture
658 the retreat culture of Confucianism: What is that? Please rephrase or explain what this is
659 prosperity. These periods saw the finalization…
667 the three cardinal system: Please explain what that was.
674 and those related to the common folk.
675 both belonged
676 the private Buddhist hall: Change to “Private Buddhist halls…were informal…
678 and were
681 The secularization…: new paragraph here
683 Holy Face Hall, Monk Hall: need Chinese terms here.
683 had ancestral pictures reflecting…
688 the differences in levels: Which levels? Official/informal? Different sects? This is unclear.
692 the temple hierarchy: change the to a
693 like sinicization, you need to define secularization when you first begin to use the term
698 monks advocate: You mean they promoted? Use past tense. Is it correct that they all promoted this?
699 of strengthening: change to: to strengthen
710 Making Liangzhe Lu the center? What does this mean? Unclear.
712 Sub-Temples
713 also began to change
722 cold spring and stream? Just say stream
726 change are to were
733 further expanded the trends of…
737 the precepts of Vinaya: Do you mean a Vinaya hall? Please clarify this sentence.
737 are included to were included
738 has also been change to was also integrated
740 coexisted
750 the natural landscape: change to natural landscapes
754 Is a collection of concrete Buddhist culture: Does not make sense. Do you mean is an expression of…?
757 affected
759 were
763 it: what is it? The study? Buddhist architecture?
766 scientific: Another term that you should define. Better to delete this term.
771 cluster
782 the mean center: not clear what you mean
810 it: what is it?
819 the popularization: eliminate the; say the trend toward popularization
820 it contributes: how?
824 in terms of the layout: eliminate the
827 the mountains: eliminate the
854 ancient Buddhist architectural sites
I notice you did not discuss architecture during the Republican period, although you included that period in your periodization.
Comments on the Quality of English Language
Most of the comments above were about English. The paper is improved but has numerous problems still.
Reviewer 3 Report
Comments and Suggestions for Authors
I can see that the author has made considerable changes with the original article. The changes include both language and content. I should say that these changes are good. However, there are still some fundamental questions needed to be fixed.
1. The author makes the following statement in the conclusion "Chinese Buddhist architecture is a collection of concrete Buddhist culture, including 754 social attributes, and represents the highest level of traditional Chinese architectural cul-755 ture. Natural factors, traffic factors, political factors and religious cultural transmission 756 factors jointly affect the continuous evolution of the spatio-temporal distribution of Bud-757 dhist architecture in Zhejiang. In particular, the secularization of Buddhism and the inte-758 gration of Confucianism, Taoism and Buddhism are the most important driving forces 759 affecting the evolution of Buddhist architectural forms. " I consider this statement as the thesis of the article. However, in the text I rarely see the analyses of the political and cultural factors. For instance, during the Song Dynasty, the population as well as the cultural center shifted from the north to the south. The capital city moved to Hangzhou. That brought tremendous changes to the Chinese society, including religion. Also, the commercialization of the society also causes the change of religion. Other factors such as technological development and education, etc., all contribute to the changes and cluster of Buddhism architectures.
2. In the original text, the author mentioned technical factor, which is a good point. However, it disappeared in the revision. Instead, the author focuses mainly on the natural factor and the influence from Confucianism. The social, political, cultural, including technical factors, which should play significant importance in the geographical cluster or changes of Buddhist architectures, all almost non-existent. That cannot justifies the key word "secluarization" for this paper.
3. The author cited someone explaining "secularization" as that "the secularization of Buddhism refers to the process of the religion's changes in the way that it interact with secular society in China, which manifests itself as the process of constantly adjusting itself in order to adapt to secular society" (pp.18). First, this explanation of secularization comes too late to the article. Secondly, what does it mean by secularization? Does it mean social, political, cultural factors involved in the Buddhist architectures? However, where are they in the text?
4. Jiangnan and Zhejiang cannot be exchanged without explanation.
5. Literature review has been improved. However, the focus should not only be on what is examined but also what is discovered.
6. The history of Buddhism and its architecture in China as well as the methodology is too long. What is unique about this article, which is the analysis of the secularization, is too short and insufficient.
Comments on the Quality of English LanguageThe English has good changes. It can be further improved.
Round 3
Reviewer 2 Report
Comments and Suggestions for Authors
Please work with a professional editor to fix the English. It still does not read well. In particular the argument is not clear. The argument should be simple and unambiguous. Some of the problem is word choice-it sounds like you are translating concepts from Chinese that are inexact in English. You should use words and phrases that are common and clear to English readers.
Comments on the Quality of English LanguageExtensive English editing still required.
Reviewer 3 Report
Comments and Suggestions for Authors
1. I see that the authors have made improvements. For instance, the authors add a section entitled "The flourishing development of Buddhist architecture under the influence of diverse societal factors." However, I find the examples given are mostly about statues rather than architecture. Also, this should be one of the sections needing elaborated on.
2. The analyses cannot explain the data convincingly. For instance, what are the commonalities behind the spatial distribution of the Buddhist architecture? What are the reasons leading to the historical changes? The authors only writer about the characteristics of Buddhist architecture one dynasty after that but fail to tell the thread of the changes over the history.
3. The conclusions come too late. The findings should have been discussed in the introductory paragraphs. Between line 1043 and 1049, the authors discuss the characteristics of "single-core cluster" and "multiple core cluster." That seems to be one of the central findings of the article. However, I do not see it developed in the essay.
Comments on the Quality of English LanguageEnglish still needs to be revised. For instance, there are no sentences from line 430 to 432.